# ReNovo: Retrieval-Based *De Novo* Mass Spectrometry Peptide Sequencing

**Shaorong Chen[1]\*, Jun Xia[2,†], Jingbo Zhou[2,\*], Lecheng Zhang[2], Zhangyang Gao[2], Bozhen Hu[2], Cheng Tan[2], Wenjie Du[3], Stan Z. Li[2†]**
[1]Zhejiang University, [2]School of Engineering, Westlake University,
[3]University of Science and Technology of China
Hangzhou, China
{chenshaorong,xiajun,stan.zq.li}@westlake.edu.cn

## Abstract

Proteomics is the large-scale study of proteins. Tandem mass spectrometry, as the only high-throughput technique for protein sequence identification, plays a pivotal role in proteomics research. One of the long-standing challenges in this field is peptide identification, which entails determining the specific peptide (sequence of amino acids) that corresponds to each observed mass spectrum. The conventional approach involves database searching, wherein the observed mass spectrum is scored against a pre-constructed peptide database. However, the reliance on pre-existing databases limits applicability in scenarios where the peptide is absent from existing databases. Such circumstances necessitate *de novo* peptide sequencing, which derives peptide sequence solely from input mass spectrum, independent of any peptide database. Despite ongoing advancements in *de novo* peptide sequencing, its performance still has considerable room for improvement, which limits its application in large-scale experiments. In this study, we introduce a novel **Re**trieval-based *De Novo* peptide sequencing methodology, termed **ReNovo**, which draws inspiration from database search methods. Specifically, by constructing a datastore from training data, ReNovo can retrieve information from the datastore during the inference stage to conduct retrieval-based inference, thereby achieving improved performance. This innovative approach enables ReNovo to effectively combine the strengths of both methods: utilizing the assistance of the datastore while also being capable of predicting novel peptides that are not present in pre-existing databases. A series of experiments have confirmed that ReNovo outperforms state-of-the-art models across multiple widely-used datasets, incurring only minor storage and time consumption, representing a significant advancement in proteomics. Supplementary materials include the code.

## 1 Introduction

In proteomics, a field dedicated to studying proteins within living organisms, mass spectrometry serves as a crucial tool protein identification. The core of this process is resolving the **peptide identification** challenge involves deciphering the amino acid sequence of a peptide from observed mass spectrum (MS) data, as illustrated in Figure 1. A standard peptide identification workflow can be delineated as follows: Initially, proteins are broken down into their constituent peptides through enzymatic digestion. The resulting peptides are then separated via liquid chromatography, producing primary scan (MS1) spectra. These spectra reveal the mass-to-charge (*m/z*) ratios of the intact peptides (also known as the **precursor**s). Following this, the peptides undergo fragmentation, yielding secondary scan (**MS2**) spectra. peptide identification involves inferring the peptide sequence from the observed mass spectrum and precursor, thus facilitating comprehensive proteomic analyses.

As depicted in Figure 1, there are currently two mainstream methods employed in peptide identification: **database search methods** and *de novo* **peptide sequencing methods**.

---

\*Equal contribution.
†Corresponding author.

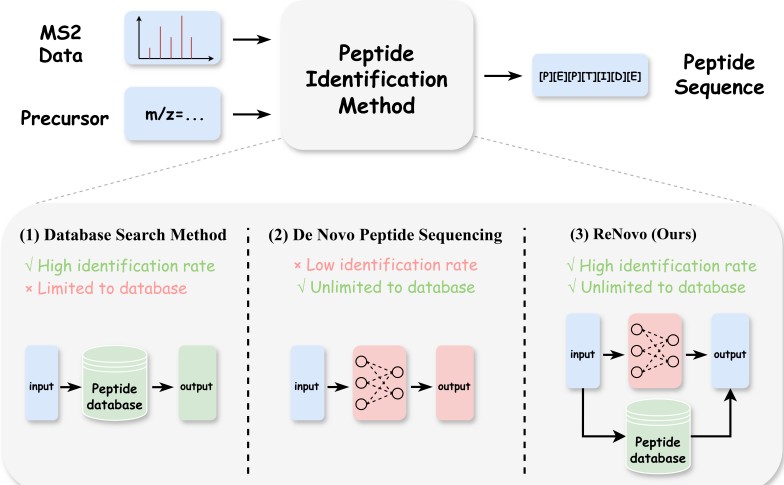

Figure 1: The semantic diagram of database search, *de novo* peptide sequencing, and ReNovo.

**Database search methods** involve comparing observed mass spectrum against a pre-established peptide database of peptide-spectrum matches (PSMs). Despite its widespread adoption, this approach has a significant limitation: constrained identification scope. The database search method inherently restricts the analysis to only those peptides present within the supplied database. In scenarios where the peptide is not present in any pre-existing database or prior knowledge of the peptide database is limited, constructing a comprehensive peptide database becomes infeasible.

*De novo* **peptide sequencing** represents an alternative method that does not rely on existing database. This method involves the direct interpretation of mass spectrum to infer peptide and is invaluable in scenarios where databases are unavailable. Such situations include but are not limited to, the sequencing of antibodies (Beslic et al., 2023), the identification of novel antigens (Karunratanakul et al., 2019), and the sequencing of metaproteomes lacking established databases (Hettich et al., 2013). However, substantial improvements are necessary to fully leverage its potential.

In this work, we have endeavored to synergize the strengths of the two aforementioned peptide identification methods by integrating concepts from database search methods into *de novo* peptide sequencing task. This innovative approach, which we term **ReNovo** (**Re**trieval-based de **Novo** peptide sequencing), is a *de novo* peptide sequencing method that is integrated with a novel retrieval-based framework. After the model training stage, ReNovo utilizes the training data to build a datastore. During the inference stage, this datastore is queried and the retrieved information will be used for the final prediction. This innovative approach enables ReNovo to effectively combine the strengths of both the database search and *de novo* peptide sequencing methods: utilizing the assistance of the datastore during the inference stage to enhance the performance, while also being able to predict novel peptide sequences that are not present in any pre-existing peptide databases. The connections and differences among these three methods can be seen in Figure 1.

Our contributions can be summarized as follows:

1. **Superior Performance.** We have developed ReNovo, an innovative *de novo* peptide sequencing method that demonstrates superior performance surpassing all the current state-of-the-art models across multiple widely-used datasets and evaluation metrics.

2. **Novel Methodology.** ReNovo utilizes an innovative "Model Training Stage - Datastore Building Stage - Retrieval Based Inference Stage" pipeline. To the best of our knowledge, ReNovo is the first-of-its-kind retrieval-based framework for *de novo* peptide sequencing task. This novel paradigm opens new avenues for proteomics researchers.

3. **Low Overhead.** Compared to other *de novo* peptide sequencing methods, ReNovo's unique datastore building stage and retrieval-based inference stage only incur minor time and storage consumption, yet result in a substantial performance improvement.

## 2 RELATED WORK

### 2.1 DATABASE SEARCH METHOD

Database search methods (Eng et al., 1994; Perkins et al., 1999; Cox & Mann, 2008) involve comparing the mass spectrum against a pre-established database. Notably, error-tolerant and open database search methods (Han et al., 2004; Renard et al., 2012; Kong et al., 2017) enable the identification of peptides with unexpected modifications or mutations. However, the effectiveness of these methods relies on a comprehensive external peptide database that encompasses all potential peptides, which limits their applicability in significant scenarios where comprehensive databases are unavailable.

### 2.2 *De Novo* PEPTIDE SEQUENCING METHOD

This method involves the direct interpretation of mass spectrum to infer peptide sequence, independent of any peptide database. Initially, researchers employed graph-based methods (Dančík et al., 1999; Taylor & Johnson, 1997), dynamic programming algorithms (Ma et al., 2003) and Hidden Markov Models (Fischer et al., 2005) to tackle the de novo peptide sequencing task. With the prosperity of deep learning, DeepNovo (Tran et al., 2017) is the first method applying deep neural networks to predict the peptide sequence. Recently, Casanovo (Yilmaz et al., 2022) first employs transformer (Vaswani, 2017) to the task of *de novo* peptide sequencing. Since then, a series of transformer-based models (Eloff et al., 2023; Yang et al., 2024; Xia et al., 2024) have been proposed to address specific challenges in *de novo* peptide sequencing. However, the performance of previous models is still far from what is required for large-scale applications in proteomics. Additionally, compared to previous models that only use the training set for model training, ReNovo significantly enhances performance by leveraging a datastore constructed from the training data.

### 2.3 RETRIEVAL-AUGMENTED GENERATION

A series of studies have widely applied retrieval mechanisms to machine translation. Notably, the kNN-LM (Khandelwal et al., 2019) and its subsequent adaptations (Khandelwal et al., 2020);(Zheng et al., 2021);(Martins et al., 2022) augment the model with a token-level symbolic database. Recently, retrieval-augmented generation (RAG) has gained success across various tasks including language modeling (Borgeaud et al., 2022);(Ram et al., 2023), question answering (Lewis et al., 2020);(Asai et al., 2023) and so on. ReNovo model draws inspiration from them. However, ReNovo is novel compared to recent advances in RAG because the datastore retrieved by ReNovo originates solely from training data, rather than from any external or domain-specific sources as in RAG. This choice aims to simulate real-world de novo applications without reliance on external databases.

## 3 METHODOLOGY

The ReNovo model undergoes three sequential stages: (1) **Model Training Stage**: During this stage, the ReNovo model is trained in a supervised manner using the training set. (2) **Datastore Building Stage**: Once the ReNovo model is trained, it generates context feature - target amino acid pairs using training data, which are then stored in the datastore. (3) **Retrieval-Based Inference Stage**: During inference, when performing next amino acid prediction, the ReNovo model will retrieve the datastore and incorporate the retrieved context feature - target amino acid pairs to make the final prediction. We will introduce the task formulation in Section 3.1, followed by detailed stage-by-stage introduction of the three stages in Section 3.2, Section 3.3 and Section 3.4, respectively.

### 3.1 TASK FORMULATION

Formally, we define the input MS2 data as: $\mathbf{s} = \{s_i\}_{i=1}^{M} = \{(m_i, I_i)\}_{i=1}^{M}$ , where each peak $s_i = (m_i, I_i)$ is a 2-tuple comprising the mass-to-charge ratio (m/z) $m_i \in \mathbb{R}$ and intensity value $I_i \in \mathbb{R}$. $M$ denotes the number of peaks, which may vary across different MS2 data. We define the precursor as $\mathbf{p} = (m_{\text{prec}}, c_{\text{prec}})$, where $m_{\text{prec}} \in \mathbb{R}$ represents the m/z value of the precursor and $c_{\text{prec}} \in \{1, 2, \ldots, 10\}$ indicates the charge state of the precursor ion. The peptide is denoted as a sequence of amino acids $\mathbf{y} = \{y_i\}_{i=1}^{N} = (y_1, y_2, \ldots, y_N)$, where $y_i \in \mathbb{AA}$ represents the identity of the $i$-th amino acid and $\mathbb{AA}$ represents the set of all considered amino acid types. $N$ corresponds

to the peptide length, which may vary among different peptides. We represent the **training set** as $(\mathcal{S}, \mathcal{P}, \mathcal{Y})$, where $\mathcal{S}$, $\mathcal{P}$ and $\mathcal{Y}$ represent the sets of MS2 data $\mathbf{s} \in \mathcal{S}$, precursor $\mathbf{p} \in \mathcal{P}$, and ground truth peptide sequences $\mathbf{y} \in \mathcal{Y}$ in the training set, respectively.

The *de novo* peptide sequencing models are designed to predict the peptide $\mathbf{y}$ given MS2 data $\mathbf{s}$ and precursor $\mathbf{p}$:

$$P(\mathbf{y} \mid \mathbf{s}, \mathbf{p}; \theta), \tag{1}$$

where $\theta$ is the parameter. If the model performs autoregressive generation, then the prediction at time step $t$ can be expressed as $p(y_t \mid y_{1:t-1}, \mathbf{s}, \mathbf{p}; \theta)$, and the above equation can be further expanded as:

$$P(\mathbf{y} \mid \mathbf{s}, \mathbf{p}; \theta) = \prod_{t=1}^{N} p(y_t \mid y_{1:t-1}, \mathbf{s}, \mathbf{p}; \theta) \tag{2}$$

## 3.2 MODEL TRAINING STAGE

As shown in Figure 2, the trainable components of ReNovo consist of a **MS2 Encoder** and a **Peptide Decoder**, all of which are based on the transformer architecture (Vaswani, 2017). In order to encode MS2 data $\mathbf{s} = \{s_i\}_{i=1}^{M}$ into feature vectors $\{E_i\}_{i=1}^{M}$, we follow previous methods (Yilmaz et al., 2022; Xia et al., 2024), treating each peak $s_i = (m_i, I_i)$ as a "word" in a MS2 "sentence" $\mathbf{s}$. The peak embedding $E_i$ is obtained by separately encoding its *m/z* value $m_i$ and intensity value $I_i$ into $E_i^m$ and $E_i^I$, then combining them through summation. Formally,

$$E_i^m = \left[\sin \frac{m_i}{N_1 N_2^{\frac{2}{d}}}, \sin \frac{m_i}{N_1 N_2^{\frac{4}{d}}}, \dots, \sin \frac{m_i}{N_1 N_2^{\frac{d}{d}}}, \cos \frac{m_i}{N_1 N_2^{\frac{d+2}{d}}}, \cos \frac{m_i}{N_1 N_2^{\frac{d+4}{d}}}, \dots, \cos \frac{m_i}{N_1 N_2^{\frac{d+d}{d}}}\right] \tag{3}$$

$$E_i^I = W I_i \tag{4}$$

$$E_i = E_i^I + E_i^m \tag{5}$$

where $d$ denotes the feature dimension, $W \in \mathbb{R}^{d \times 1}$ represents a trainable linear layer, $N_1$ and $N_2$ are user-defined scalars that can be set to any value. Specifically, we set $d = 512$, $N_1 = \frac{m_{\max}}{m_{\min}}$ and $N_2 = \frac{m_{\min}}{2\pi}$, where $m_{\max} = 10,000$ and $m_{\min} = 0.001$ in our work. These features provide a detailed representation of *m/z* value and help the model attend to *m/z* differences between peaks.

After encoding each peak $s_i$ of the input MS2 data $\mathbf{s} = \{s_i\}_{i=1}^{M} = \{s_1, \dots, s_M\}$ to obtain the features $\{E_i\}_{i=1}^{M} = \{E_1, \dots, E_M\}$, these feature sequences are then fed into the MS2 Encoder. The Peptide Decoder generates the sequence $\mathbf{y}$ autoregressively, producing each amino acid $y_t$ through $p(y_t \mid y_{1:t-1}, \mathbf{s}, \mathbf{p}; \theta)$ step by step based on the previously predicted amino acids $y_{1:t-1}$. The method for encoding precursors $\mathbf{p} = (m_{\text{prec}}, c_{\text{prec}})$ is consistent with the method used for encoding peaks. During the training stage, ReNovo employs teacher-forcing strategy Williams & Zipser (1989), where $y_{1:t-1}$ corresponds to the ground truth peptide $\mathbf{y} \in \mathcal{Y}$. The loss function used for training ReNovo is cross-entropy loss:

$$\mathcal{L}(\theta) = -\sum_{t=1}^{N} \log p(\hat{y}_t \mid y_{1:t-1}, \mathbf{s}, \mathbf{p}; \theta) \tag{6}$$

where $\theta$ represents all the trainable parameters of the ReNovo model, $\hat{y}_t$ represents the ground truth target amino acid in $\mathbf{y} \in \mathcal{Y}$.

## 3.3 DATASTORE BUILDING STAGE

In this section, we will first define the "context feature" and the "target amino acid". Specifically, the ReNovo's prediction at time step $t$ can be expressed as Equation 7. We refer to $f(y_{1:t-1}, \mathbf{s}, \mathbf{p}; \theta^*)$ output by the Peptide Decoder of ReNovo as the **context feature**, a high-dimensional feature vector. $y_t$ is the **target amino acid** corresponding to the context feature $f(y_{1:t-1}, \mathbf{s}, \mathbf{p}; \theta^*)$.

$$p(y_t \mid y_{1:t-1}, \mathbf{s}, \mathbf{p}; \theta^*) = p(y_t \mid f(y_{1:t-1}, \mathbf{s}, \mathbf{p}; \theta^*)) \tag{7}$$

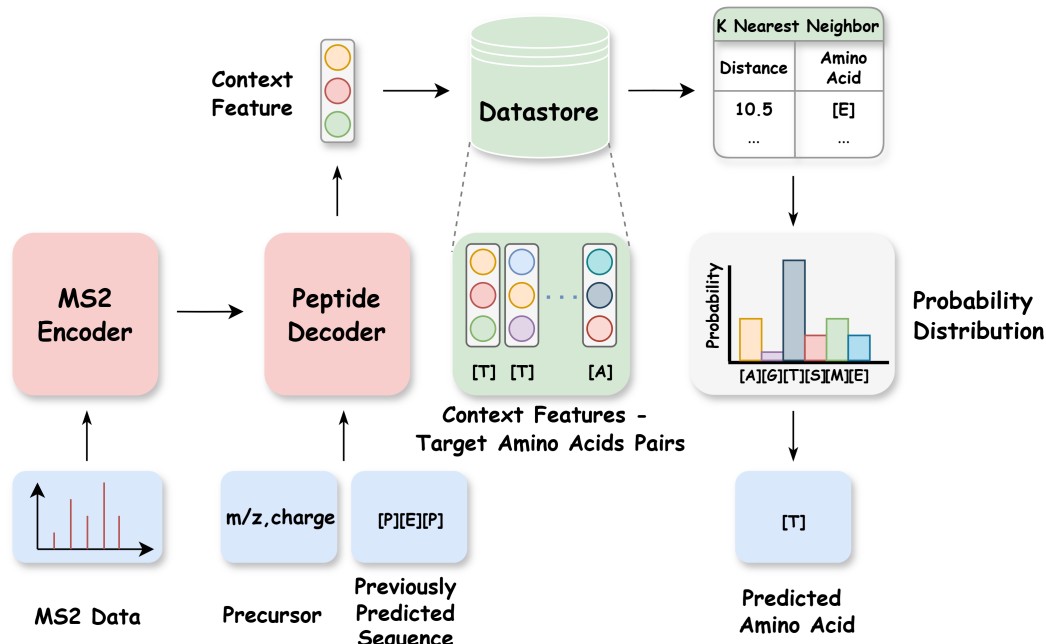

Figure 2: The framework of the ReNovo model. The ReNovo model takes MS2 data, the precursor and the previously predicted sequence as inputs, and produces a context feature as output. This context feature is then used to query the datastore to yield the final predicted amino acid.

After the model training stage, ReNovo utilizes the pre-trained model $P(\mathbf{y} \mid \mathbf{s}, \mathbf{p}; \theta^*)$ along with the training dataset $(\mathcal{S}, \mathcal{P}, \mathcal{Y})$ to build the datastore $\mathcal{D}$, where $\theta^*$ denotes the parameters of the pre-trained ReNovo model. Formally, the datastore $\mathcal{D}$ can be represented as:

$$\mathcal{D} = (\mathcal{K}, \mathcal{V}) = \{(f(y_{1:t-1}, \mathbf{s}, \mathbf{p}; \theta^*), y_t), \forall y_t \in \mathbf{y} \mid \forall (\mathbf{s}, \mathbf{p}, \mathbf{y}) \in (\mathcal{S}, \mathcal{P}, \mathcal{Y})\} \qquad (8)$$

In other words, the key-value pairs $(\mathcal{K}, \mathcal{V})$ stored in the datastore $\mathcal{D}$ can be interpreted as **context feature - target amino acid pairs** $(f(y_{1:t-1}, \mathbf{s}, \mathbf{p}; \theta^*), y_t)$. The context feature $f(y_{1:t-1}, \mathbf{s}, \mathbf{p}; \theta^*)$ is a feature vector generated by the pre-trained ReNovo model, which encapsulates information from the input MS2 spectra $\mathbf{s}$, precursor $\mathbf{p}$, and the previously predicted peptide sequence $y_{1:t-1}$. The target amino acid $y_t$ represents the ground truth prediction that the ReNovo model should output under the corresponding context represented by $f(y_{1:t-1}, \mathbf{s}, \mathbf{p}; \theta^*)$. By constructing such a context feature - target amino acid pairs datastore $\mathcal{D}$, ReNovo essentially "memorizes" which target amino acid $y_t$ to output under various contextual conditions $f(y_{1:t-1}, \mathbf{s}, \mathbf{p}; \theta^*)$. Consequently, during the retrieval-based inference stage which will be discussed in the next section, ReNovo can leverage the knowledge stored in the datastore $\mathcal{D}$ to make more accurate predictions.

To distinguish the ReNovo model from database search methods, we refer to the database used by the ReNovo model as the "datastore". The datastore of the ReNovo is constructed using the data that is used to train the ReNovo model, thus making it inherently a *de novo* peptide sequencing model. This paradigm ensures a fair performance comparison with other *de novo* peptide sequencing methods. In contrast, database used in database search methods are built from external data sources.

### 3.4 RETRIEVAL-BASED INFERENCE STAGE

During the inference stage, the model can be represented by a schematic diagram in Figure 2. ReNovo augments the pre-trained model with a retrieval mechanism that enables direct access to a datastore $\mathcal{D} = (\mathcal{K}, \mathcal{V})$ during the inference stage. We refer to this inference approach as **Retrieval-Based Inference**. During the retrieval-based inference stage, given MS2 spectrum $\mathbf{s}$ and precursor $\mathbf{p}$ in the test dataset, the model generates the sequence $\mathbf{y}$ autoregressively, producing each amino acid step

by step. At time step $t$, the ReNovo model first produces the context feature $f(y_{1:t-1}, \mathbf{s}, \mathbf{p}; \theta^*)$. This context feature is then utilized to query the datastore $\mathcal{D}$ to find the $K$ nearest neighbors $\mathcal{N} = \{(k_j, v_j), j \in (1, 2, \ldots, K)\}$, where $K$ is an integer hyperparameter and $(k_j, v_j)$ represent the context feature - target amino acid pairs retrieved with the Euclidean distances.

The retrieved context feature - target amino acid pairs $\mathcal{N} = \{(k_j, v_j), j \in (1, 2, \ldots, K)\}$ are subsequently transformed into a probability distribution over the amino acids vocabulary $\mathbb{AA}$ by:

$$p_{\text{kNN}}(y_t \mid y_{1:t-1}, \mathbf{s}, \mathbf{p}; \theta^*) = \sum_{(k_j, v_j) \in \mathcal{N}} \mathbb{1}_{y_t = v_j} \exp\left(\frac{-d(k_j, f(y_{1:t-1}, \mathbf{s}, \mathbf{p}; \theta^*))}{T}\right), y_t \in \mathbb{AA} \quad (9)$$

$$p_{\text{aa}}(y_t \mid y_{1:t-1}, \mathbf{s}, \mathbf{p}; \theta^*) = \frac{p_{\text{kNN}}(y_t \mid y_{1:t-1}, \mathbf{s}, \mathbf{p}; \theta^*)}{\sum_{y_j \in \mathbb{AA}} p_{\text{kNN}}(y_j \mid y_{1:t-1}, \mathbf{s}, \mathbf{p}; \theta^*)}, y_t \in \mathbb{AA} \quad (10)$$

where $d(k_j, f(y_{1:t-1}, \mathbf{s}, \mathbf{p}; \theta^*))$ denotes the Euclidean distance between the query context feature $f(y_{1:t-1}, \mathbf{s}, \mathbf{p}; \theta^*)$ and the retrieved context feature $k_j$, and $T$ denotes the temperature that is used to control the smoothness of the probabilities. The resulting $p_{\text{aa}}$ can be interpreted as the probability generated by ReNovo with the datastore retrieval, which is used to predict amino acid $y_t$.

# 4 EXPERIMENTS

We have chosen to use several advanced and representative models (Deepnovo (Tran et al., 2017), Pointnovo (Qiao et al., 2021), Casanovo (Yilmaz et al., 2022), Instanovo (Eloff et al., 2023), Helixnovo (Yang et al., 2024), Adanovo (Xia et al., 2024)) as baselines for comparison with ReNovo. This setup ensures a thorough and fair assessment of our ReNovo's performance against a range of current advanced models. To maintain consistency and facilitate direct comparisons, we have standardized the experimental settings across all models unless indicated otherwise.

## 4.1 DATASET AND EVALUATION METRICS

For evaluation, we utilize three representative datasets: **Seven-species Dataset** (Tran et al., 2017), **Nine-species Dataset**(Tran et al., 2017), and **HC-PT Dataset**(Eloff et al., 2023). These datasets were selected due to their varying sizes and characteristics in terms of resolution and origins. A detailed description of the dataset and analysis of dataset overlap is provided in Appendix A.

The evaluation metrics we considered include: (1) **Peptide-level precision**. This metric acts as the primary measure of a model's practical effectiveness. It is defined as the proportion of peptides correctly predicted by the model. (2) **Peptide-level AUC**. It is determined by calculating the area under the peptide-level precision-recall curve, providing a comprehensive evaluation. (3) **Amino acid-level precision and recall**. These metrics offer a more granular assessment of model performance at the individual amino acid level. They are defined as the precision and recall of predictions across all amino acids in the test set. Further details on can be found in Appendix B.

## 4.2 EXPERIMENT SETUP

To simulate real-world scenarios requiring novel peptide identification, we utilized a leave-one-out approach, akin to prior *de novo* peptide sequencing studies. For example, all the models were trained on data from six species and subsequently tested on the remaining species in Seven-species Dataset. The same applies to the other datasets. In the experimental section, we do not compare performance with database search methods. On one hand, ReNovo is inherently a *de novo* method, thus comparisons with similar *de novo* models are more appropriate. On the other hand, we employ a leave-one-out methodology, ensuring that peptides appearing in the testing set are completely distinct from those in the training set, a scenario where database search methods are not applicable.

## 4.3 MAIN RESULTS

The main results are shown in Table 1. We compared ReNovo with representative baseline models across three datasets and four evaluation metrics, as previously mentioned.

Table 1: Empirical comparison of *de novo* peptide sequencing models. The best and the second best are highlighted with **bold** and underline, respectively.

| Method | Peptide-level performance | | | | | | Amino acid-level performance | | | | | |
| --- | --- | --- | --- | --- | --- | --- | --- | --- | --- | --- | --- | --- |
| | Seven-species | | Nine-species | | HC-PT | | Seven-species | | Nine-species | | HC-PT | |
| | Prec. | AUC | Prec. | AUC | Prec. | AUC | Prec. | Recall | Prec. | Recall | Prec. | Recall |
| DeepNovo | 0.204 | 0.136 | 0.428 | 0.376 | 0.313 | 0.255 | 0.492 | 0.433 | 0.696 | 0.638 | 0.531 | 0.534 |
| PointNovo | 0.022 | 0.007 | 0.480 | 0.436 | 0.419 | 0.373 | 0.196 | 0.169 | 0.740 | 0.671 | 0.623 | 0.622 |
| CasaNovo | 0.119 | 0.084 | 0.481 | 0.439 | 0.211 | 0.177 | 0.322 | 0.327 | 0.697 | 0.696 | 0.442 | 0.453 |
| HelixNovo | 0.234 | 0.173 | 0.517 | 0.453 | 0.356 | 0.318 | 0.481 | 0.472 | 0.765 | 0.758 | 0.588 | 0.582 |
| AdaNovo | 0.174 | 0.135 | 0.505 | 0.469 | 0.212 | 0.178 | 0.379 | 0.385 | 0.698 | 0.709 | 0.442 | 0.451 |
| **ReNovo** | **0.278** | **0.228** | **0.568** | **0.528** | **0.467** | **0.436** | **0.512** | **0.514** | **0.770** | **0.769** | **0.651** | **0.648** |

**ReNovo Outperforms All Current State-Of-The-Art Models Across All Datasets**

As shown in Table 1, ReNovo surpasses all the baseline models in both peptide-level evaluation metrics and amino acid-level evaluation metrics across all three datasets.

In terms of the most critical metric, peptide-level precision, ReNovo outperforms the best baseline models (HelixNovo and PointNovo) by 18.8%, 9.86%, and 11.46% across the three datasets, respectively, achieving an average improvement of 13.31%. This highlights that ReNovo's improvements are both substantial and consistent. Additionally, ReNovo demonstrates remarkable results in peptide-level AUC, surpassing the top baseline models (HelixNovo, AdaNovo and PointNovo) by 31.79%, 12.58%, and 16.89% across the there datasets, respectively, with an average increase of 20.42%. Peptide-level AUC offers a comprehensive evaluation of model performance, further reinforcing that ReNovo's performance enhancements are both significant and well-rounded.

At the amino acid-level, ReNovo also exhibits notable improvements. Across the three datasets, ReNovo improves amino acid precision over the best baseline models by 4.07%, 0.65%, and 5.49%, respectively. Similarly, its amino acid recall increases by 8.9%, 1.45%, and 4.18%, respectively. An average improvement of 4.12% was observed across all amino acid-level metrics.

**ReNovo Achieves More Significant Improvement in Peptide-level Metrics.**

It is noteworthy that despite ReNovo utilizing amino acid-level retrieval where the next amino acid $y_t$ is predicted based on the context feature - target amino acid pairs retrieved from the datastore $\mathcal{D}$, the gains at the peptide level (13.31% in precision and 20.42% in AUC) are significantly higher than those at the amino acid level (4.12%). This indicates that the information retrieved from the datastore pertains to the entire peptide sequence rather than being confined to individual amino acid-level information, enabling ReNovo to achieve superior performance at the peptide level performance, which is a more crucial factor when evaluating a *de novo* peptide sequencing model.

## 4.4 ABLATION STUDY AND HYPERPARAMETER TUNNING

**Ablation Study: The Retrieval Mechanism of ReNovo Contributes Substantially.**

We conducted ablation experiments to evaluate three different configurations: (1) **ReNovo w/o Retrieval**: using the logits from the Peptide Decoder without relying on datastore retrieval, (2) **ReNovo**: using only the prediction results retrieved from the datastore, (3) **ReNovo with Residual**: combining both (1) and (2) via a residual connection. We analyzed the impact of these configurations on ReNovo's performance, with the results shown in Figure 3. We observed a significant performance decline in the ReNovo model without datastore retrieval (ReNovo w/o retrieval), which demonstrates that the context feature - target amino acid pairs retrieved from the datastore contribute substantially to ReNovo's performance improvements and indicates that the retrieval-based framework of ReNovo is highly effective. Although a pure retrieval approach (**ReNovo**) is effective, we further improve performance by combining the probability distribution obtained from using the logits from the Peptide Decoder with the probability distribution in Equation 10 based on retrieval via a residual connection. This configuration (**ReNovo with Residual**) yields a modest performance gain.

**ReNovo's Robustness With Respect To the Number of Retrieved Pairs $K$**

During the retrieval-based inference stage, the ReNovo model retrieves $K$ context feature - target amino acid pairs from the datastore $\mathcal{D}$. A larger $K$ means the ReNovo model retrieves more pairs.

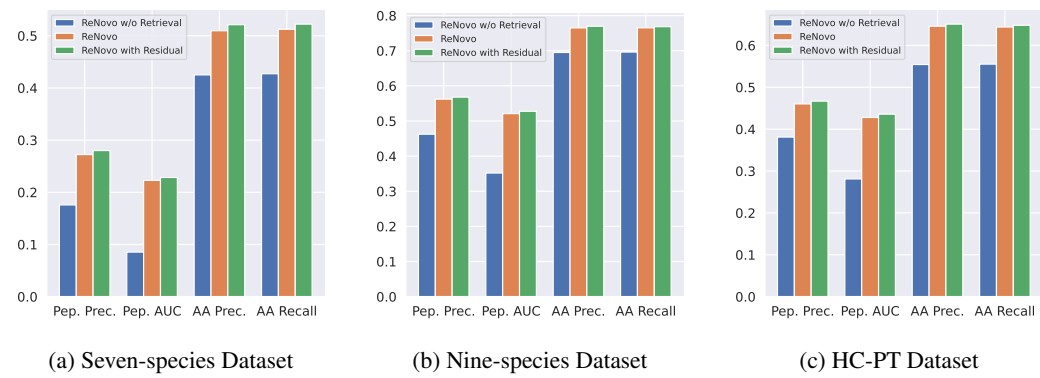

(a) Seven-species Dataset  (b) Nine-species Dataset  (c) HC-PT Dataset

Figure 3: ReNovo's ablation experiments conducted on three datasets.

We conducted sensitivity analysis experiments on the Seven-species Dataset to investigate the impact of the parameter $K$ on the ReNovo's performance, and the results are shown in Figure 4. We observed that when $K$ is relatively small ($K < 32$), increasing $K$ results in significant performance improvement because the ReNovo model can more effectively leverage the information available in the datastore. However, further increases in $K$ provide minimal additional performance gains.

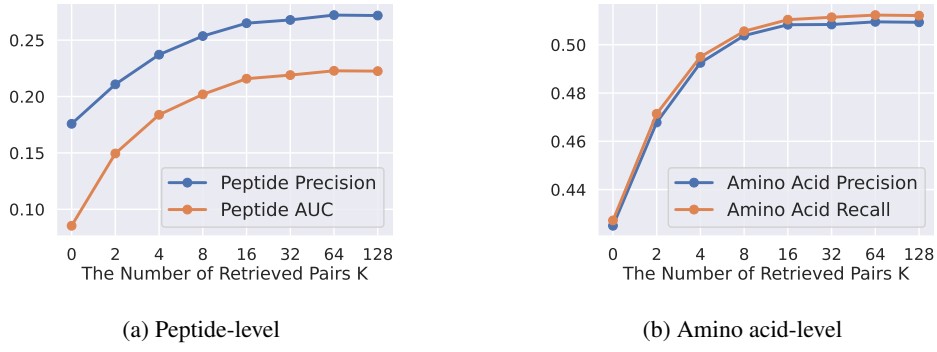

(a) Peptide-level  (b) Amino acid-level

Figure 4: The impact of the number of pairs $K$ on the performance of the ReNovo model.

**ReNovo's Robustnes With Respect To the Temperature $T$**

In Equation 10, the temperature parameter $T$ controls the smoothness of the output probabilities. We conducted sensitivity analysis on the Seven-species Dataset to investigate the impact of the parameter $T$ on the ReNovo's performance. The results are shown in Figure 5. We can observe that when $T$ is set within a reasonable range ($2 \sim 7$), its impact on performance is minimal.

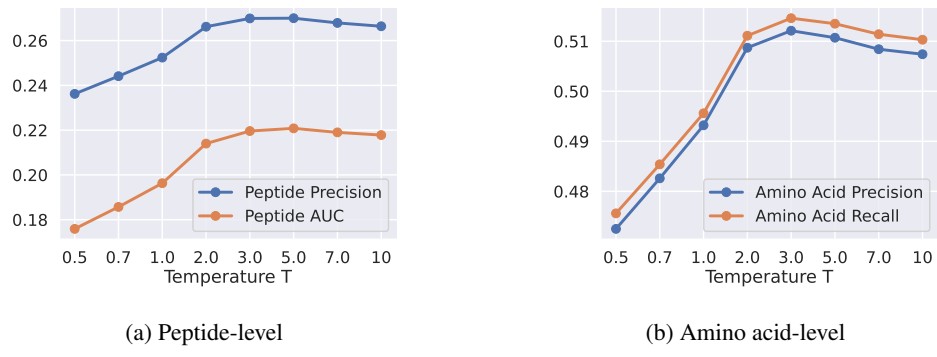

(a) Peptide-level  (b) Amino acid-level

Figure 5: The impact of the temperature $T$ on the performance of the ReNovo model.

4.5 CONSUMPTION OF TIME AND STORAGE

In this section, we examine the time and storage consumption of the ReNovo model.

**The Time Consumption of the ReNovo Model Is Negligible.**

When evaluating running times, we ensured consistent experimental setups across all models including: the use of an Nvidia A100 GPU (80GB), setting the batch size to 32, and calculating the average time by dividing the total time by the number of steps. The results, shown in Table 2, reveal minimal differences in training time between ReNovo and other models. However, ReNovo's inference stage requires datastore retrieval, resulting in longer inference time that is 1.53 to 2 times that of HelixNovo, depending on the dataset. Nevertheless, considering the substantial improvements detailed in Table 1, this increase in inference time is negligible and a reasonable trade-off.

Table 2: Training and inference times of various models.

| Model | Training Time (s) | | | Inference Time (s) | | |
|---|---|---|---|---|---|---|
| | 7-species | 9-species | HC-PT | 7-species | 9-species | HC-PT |
| DeepNovo | 0.31 | 0.38 | 0.30 | 0.04 | 0.07 | 0.02 |
| PointNovo | 0.34 | 0.31 | 0.28 | 0.25 | 0.24 | 0.22 |
| CasaNovo | 0.36 | 0.33 | 0.32 | 0.27 | 0.28 | 0.26 |
| InstaNovo | 0.11 | 0.10 | 0.14 | 0.03 | 0.03 | 0.04 |
| AdaNovo | 1.16 | 1.07 | 0.96 | 1.48 | 1.50 | 1.46 |
| HelixNovo | 0.56 | 0.35 | 0.41 | 0.30 | 0.28 | 0.17 |
| **ReNovo** | 0.18 | 0.11 | 0.12 | 0.46 | 0.55 | 0.34 |

Table 3: The time consumption of ReNovo at each stage.

| | Model Training | Datastore Building | Retrieval-Based Inference |
|---|---|---|---|
| Time(s) | 84,703 | 2,207 | 8,278 |
| Percentage | 88.98% | 2.32% | 8.70% |

We also compared the time consumption of the ReNovo model across three stages. The results, presented in Table 3, show that the datastore building stage accounts for approximately 2.32% of the total time in the 'Model Training - Datastore Building - Retrieval Based Inference' pipeline. Additionally, the relatively more time-consuming retrieval-based inference stage constitutes only about 8.70% of the total time for the entire pipeline. Therefore, the time consumption associated with both datastore building and retrieval-based inference is negligible.

**The Storage Consumption of the ReNovo Model Is Negligible.**

The storage consumption of ReNovo's datastore is shown in Table 4. Depending on the scales of datasets, the datastores built for the three datasets contain approximately 5.6 million, 8.4 million, and 3.2 million context feature-target amino acid pairs, respectively. The corresponding storage space occupied is 11 GB, 16 GB, and 6 GB. Considering that the datastore is plug-and-play, occupying only storage space rather than memory space, the consumption can be considered negligible.

Table 4: The datastore statistic.

| | Seven-species Dataset | Nine-species Dataset | HC-PT Dataset |
|---|---|---|---|
| Pairs Number | 5,626,944 | 8,456,240 | 3,232,616 |
| Storage (GB) | 11.16 | 16.77 | 6.42 |

4.6 CASE STUDY

Through the case study presented in Table 5, we illustrated how ReNovo, with the assistance of the datastore, can make more accurate predictions during retrieval-based inference. Given input MS2

data **s**, ReNovo autoregressively generated the peptide sequence $y_{1:t-1}$ = "RVNLARIDNE". For the next amino acid prediction for $y_t$, if the ReNovo does not leverage datastore retrieval ($K = 0$), it predicts the amino acid 'E' based on the logits ($p(\text{E}) > p(\text{D})$) output by the ReNovo's Peptide Decoder. In contrast, when employing retrieval-based inference ($K = 32$), both amino acids 'E' and 'D' are present among the target amino acids retrieved by ReNovo. After applying the weighting calculation based on Equation 9 and Equation 10, the model assigns a higher weight to amino acid 'D', resulting in the final prediction of 'D' rather than 'E'. This enabled the ReNovo, with the assistance of the datastore information, to correctly predict the entire peptide sequence. This case study illustrates that the knowledge does not need to be implicitly stored within the parameters of ReNovo. Instead, it can be explicitly acquired through a plug-and-play established datastore. By leveraging an established datastore, the accuracy of peptide predictions can be significantly enhanced.

Table 5: Case Study. Visualization can be found in Figure 7 in Appendix E

| Input MS2 | Previously Predicted | | | |
|---|---|---|---|---|
|  | RVNLARIDNE | | | |
| | **Sample 1** | **Sample 2** | **Sample 3** | **...** |
| **Retrieved MS2** |  |  |  | ... |
| **Retrieved Amino Acid** | D | E | D | ... |
| **Retrieved Distance** | 22.11 | 26.21 | 26.23 | ... |
| **Predicted Amino Acid**($K = 0$) | $p(\text{D}) = 0.45, p(\text{E}) = 0.50, ...$ | | | |
| **Predicted Amino Acid**($K = 32$) | $p(\text{D}) = 0.48, p(\text{E}) = 0.47, ...$ | | | |
| **Ground Truth Amino Acid** | D | | | |
| **Predicted Peptide**($K = 0$) | RVNLARIDNEEVM(+15.99) | | | |
| **Predicted Peptide**($K = 32$) | RVNLARIDNEDVM(+15.99) | | | |
| **Ground Truth Peptide** | RVNLARIDNEDVM(+15.99) | | | |

## 5 CONCLUSION

In this study, we introduce ReNovo, a first-of-its-kind retrieval-based *de novo* peptide sequencing model. ReNovo employs a innovative "Model Training - Datastore Building - Retrieval Based Inference" pipeline. By constructing a datastore from the training data, the ReNovo model can utilize the datastore to conduct retrieval-based inference, thereby achieving improved performance while also being able to predict novel peptide that are not present in any pre-existing peptide databases. Experiments confirm that ReNovo outperforms the state-of-the-art models across three widely-used datasets. Additionally, the time and storage consumption associated by the ReNovo are negligible. We believe that retrieval-based *de novo* peptide sequencing will establish a new paradigm in this field, offering valuable insights and inspiration to researchers in this domain. In the future, ReNovo is expected to lead to practical applications in proteomics, including enhancing the analysis of immunopeptidomics and metaproteomics, as well as enabling deeper exploration of the dark proteome.

ACKNOWLEDGMENTS

This work was supported by National Natural Science Foundation of China Project No.623B2086 and No. U21A20427, the Science & Technology Innovation 2030 Major Program Project No.2021ZD0150100, Project No.WU2022A009 from the Center of Synthetic Biology and Integrated Bioengineering of Westlake University, and Project No.WU2023C019 from the Westlake University Industries of the Future Research. Finally, we thank the Westlake University HPC Center for providing part of the computational resource.

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

## A   DATASET

### A.1   DATASET STATISTICS AND DESCRIPTION

The detailed statistics of the datasets are shown in Table 6

Table 6: The datasets statistics.

| Dataset | Avg. peaks | Avg. peptide length | train/valid/test set size |
|---|---|---|---|
| Seven-species Dataset | 466.05 | 15.79 | 317,009 / 17,740 / 17,049 |
| Nine-species Dataset | 134.91 | 15.01 | 499,402 / 28,572 / 27,142 |
| HC-PT Dataset | 184.21 | 12.53 | 213,284 / 25,718 / 26,536 |

**Seven-species Dataset** (Tran et al., 2017) This dataset comprises low-resolution mass spectra and corresponding peptide labels derived from seven distinct species. To maintain consistency with the primary experiments, our evaluation focused on the yeast species.

**Nine-species Dataset** (Tran et al., 2017) This dataset, which has been extensively utilized in previous research, contains high-resolution mass spectra and peptide labels from nine different species. It is worth noting that this dataset incorporates three post-translational modifications (PTMs): oxidation of methionine, deamidation of asparagine and deamidation of glutamine.

**HC-PT Dataset** As described in the InstaNovo (Eloff et al., 2023), the HC-PT dataset consists of synthetic tryptic peptides representing all canonical human proteins and their isoforms. It also includes peptides generated by alternative proteases and HLA peptides. The distinguishing feature of this dataset is its high-resolution spectra for human-origin peptides.

### A.2   ANALYSIS OF DATASET OVERLAP

To prevent data leakage and demonstrate that the performance improvement of the ReNovo model arises from generalization rather than overfitting to the training dataset through retrieval, we analyzed the overlap of spectra and peptides between the training and test sets across three datasets.

**For spectra overlap**, we referred to the method in Wan et al. (2002): Specifically, all spectra were discretized into binned spectra. We then calculated the average similarity between each binned spectrum in the test set and all those in the training set. Finally, we averaged these similarity values across all spectra in the testing set to obtain the overall dataset similarity (ranging from [-1, 1]), which measures the overlap between the training and test spectra. The results are shown in Table 7.

| | Seven-species Dataset | Nine-species Dataset | HC-PT Dataset |
|---|---|---|---|
| Similarity | 0.172 | 0.101 | 0.119 |

Table 7: Similarity of binned spectra between test set and training set.

From Table 7, it can be observed that the spectra similarity for all datasets is approximately 0.1. This indicates that the overlap between the spectra in the training set and the test set is quite low.

**For peptide overlap**, we employ a leave-one-out cross-validation framework in which the peptides in the training set are completely disjoint from those in the test set. In other words, during the pre-processing phase of the dataset, we ensured that there was no duplication between peptide sequences in the training and test sets, making the dataset suitable for evaluating de novo sequencing methods.

To further explore the model's performance on test sequences that lack similar sequences in the training set, we conducted experiments on the Nine-species Dataset, using AdaNovo (the best baseline for this dataset in Table 1) for comparison. We began by calculating the Levenshtein distance (Levenshtein, 1966) (a widely adopted method for measuring the similarity between two strings) for each test sequence against all sequences in the training set. We then identified the minimum Levenshtein distance for each test sequence and filtered out test sequences with a minimum Levenshtein distance <3 and <5, respectively, resulting in Nine-species Test Dataset (>3) and Nine-species Test Dataset (>5). These datasets, having excluded peptide sequences similar to those in the training set. We maintained all other experimental settings unchanged, and the results are presented in Table 8.

| Model Name | Test Set | Peptide Precision | Peptide AUC |
|---|---|---|---|
| ReNovo | Nine-species Dataset (Original) | 0.568 | 0.528 |
| AdaNovo | Nine-species Dataset (Original) | 0.505 | 0.469 |
| ReNovo | Nine-species Test Dataset (>3) | 0.465 | 0.423 |
| AdaNovo | Nine-species Test Dataset (>3) | 0.398 | 0.356 |
| ReNovo | Nine-species Test Dataset (>5) | 0.396 | 0.351 |
| AdaNovo | Nine-species Test Dataset (>5) | 0.333 | 0.292 |

Table 8: Performance Comparison on Filtered Test Datasets.

From Table 8, we observe that the performance of both ReNovo and AdaNovo declines on the filtered test datasets, which is expected as the test set contains fewer similar data to the training set. However, it is also evident that ReNovo still outperforms the state-of-the-art baseline models significantly on the same filtered test sets. This clearly indicates that the improved performance of ReNovo is due to generalization rather than overfitting to the training dataset.

## B EVALUATION METRICS

This section delineates the evaluation metrics employed in our experiment:

**Peptide-level Precision** This metric serves as the principal indicator of a model's practical utility, as the ultimate goal of *de novo* peptide sequencing is to assign complete and accurate peptide sequences to each mass spectrum. A predicted peptide $\mathbf{y}$ is considered correct only if its entire amino acid sequence matches the ground truth $\hat{\mathbf{y}}$, that is, $\mathbf{y} = \hat{\mathbf{y}}$. Given a dataset of $N_{all}^{p}$ spectra, if a model correctly predicts $N_{match}^{p}$ peptides, the peptide-level precision is calculated as $N_{match}^{p}/N_{all}^{p}$.

**Area Under the peptide-level Precision-Recall Curve (AUC)** In the following sections, we will refer to this metric as the "AUC" for simplicity. The calculation process is divided into several steps: Calculate confidence scores $sc(y_i)$ for individual amino acids $y_i$ in the ReNovo model's output $\mathbf{y}$. Define the overall confidence score of $\mathbf{y}$ as the mean of its constituent amino acid confidence scores, that is $sc(\mathbf{y}) = mean(sc(y_i)), y_i \in \mathbf{y}$. Sort predictions in descending order based on confidence scores. Starting from the highest confidence prediction, accumulate the model's peptide recall and precision values. These cumulative values are then used as the horizontal axis and the vertical axis, respectively, for the points on the precision-recall curve. The AUC value is obtained by calculating the area under the curve formed by this curve. The resulting AUC provides a robust measure of model performance, accounting for both precision and recall across different confidence levels.

**Amino Acid-level Precision and Recall** These metrics offer a more granular assessment of model performance at the individual amino acid level. The process involves: Determining the number of matched amino acid predictions $N_{match}^{aa}$ based on two criteria: a) Mass difference $< 0.1$ Da from the corresponding ground truth amino acid. b) Either prefix or suffix with a mass difference $\leq 0.5$ Da from the corresponding sequence in the ground truth peptide. Calculating amino acid-level precision and recall as $N_{match}^{aa}/N_{pred}^{aa}$ and $N_{match}^{aa}/N_{truth}^{aa}$, where $N_{pred}^{aa}$ and $N_{truth}^{aa}$ is the total number of predicted amino acids and ground truth amino acids respectively. These metrics complement the peptide-level assessments.

## C  RELIABILITY ANALYSIS

To evaluate the reliability of the ReNovo model, we included a recall-coverage curve shown in Figure 6, which can provide valuable insights into its performance across different confidence levels. Specifically, amino acid-level confidence scores are computed by applying a softmax layer to the raw output of the Peptide Decoder. We use the mean score across all amino acids to derive a peptide-level confidence score. When plotting the recall-coverage curve, all predicted sequences are ranked based on their confidence scores. For amino acid-level curves, all amino acids within a given peptide are assigned the same score.

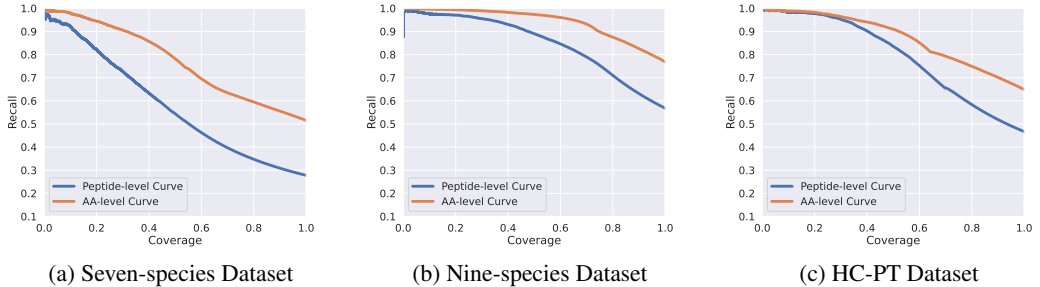

    (a) Seven-species Dataset       (b) Nine-species Dataset       (c) HC-PT Dataset

Figure 6: The recall-coverage curves for ReNovo illustrate the relationship between recall and coverage. The horizontal axis indicates coverage, while the vertical axis shows peptide-level and amino acid (AA)-level recall. The blue line corresponds to the peptide-level recall performance, whereas the yellow line represents the AA-level recall performance.

## D  VISUALIZATION OF CASE STUDY

To provide a easily interpretable visual comparison of the similarities between the retrieved context features and those generated by ReNovo for the case study in Section 4.6, we applied t-SNE dimensionality reduction Van der Maaten & Hinton (2008) to project the context features onto a two-dimensional plane. As highlighted in the case study described in Table 5, without retrieval, the model incorrectly predicts the amino acid 'E'. However, with retrieval, the ReNovo model retrieves two possible target amino acids: 'D' and 'E', represented by blue and orange color in the Figure 7, respectively. Notably, the cluster corresponding to amino acid 'D' is larger and carries greater weight. By applying the weighting calculations outlined in Equations 9 and 10, the model assigns a higher weight to amino acid 'D', resulting in a final prediction of 'D' instead of 'E'. This demonstrates how ReNovo effectively utilizes datastore to accurately predict the entire peptide sequence.

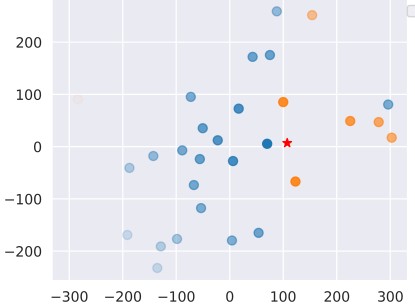

Figure 7: Visualization of Case Study. Only the 30 retrieved context features with the smallest Euclidean distance were retained and projected into a 2D space. The red pentagram represents the context feature generated by ReNovo, while the blue and orange dots correspond to the retrieved context features associated with two target amino acids: 'D' and 'E', respectively. The intensity of the colors reflects the Euclidean distance, with darker shades indicating smaller distances.

# E    THE EFFECT OF DATASTORE SIZE ON RENOVO'S PERFORMANCE

| Datastore Pairs | Datastore Size | Peptide Precision | Peptide AUC | AA Precision | AA Recall |
|---|---|---|---|---|---|
| 8,456,240 | 16.77 GB | 0.568 | 0.528 | 0.770 | 0.769 |
| 22,646,374 | 44.89 GB | 0.696 | 0.656 | 0.836 | 0.834 |

Table 9: The performance of the ReNovo model across various datastore sizes.

To further investigate the effect of datastore size on ReNovo's performance, we conducted additional experiments using a larger dataset (beyond the training set). Specifically, we utilized the ReNovo model trained on the Nine-species Dataset, along with a larger dataset from (Tran et al., 2017) that includes the same species, excluding data belonging to the same species as the test set to construct the datastore. The training set, test set, and other configurations were kept unchanged. The sizes of the two datastores and the corresponding ReNovo performance metrics are detailed in Table 9.

We find it quite surprising that adding additional data to the datastore led to such a significant performance gain, while keeping the training dataset and model weights unchanged. This experiment highlights the effectiveness of the retrieval-based mechanism: (1) Knowledge does not need to be implicitly stored within the model parameters; instead, it is explicitly stored in the datastore, which can be accessed during the inference phase in a plug-and-play manner, thereby enhancing the model's generative capabilities. (2) By generating sequences based on retrieved references rather than from scratch, the ReNovo model reduces the inherent complexity of producing coherent and contextually accurate sequences. We believe this insight could be highly enlightening for researchers in the field of peptide identification, specifically by demonstrating the potential of utilizing retrieval-based mechanisms and a large-scale datastore to improve sequencing performance.

