# OpenReview forum: "ReNovo: Retrieval-Based \emph{De Novo} Mass Spectrometry Peptide Sequencing"
_ICLR.cc/2025/Conference — ICLR 2025 Poster_

### Official Review · Reviewer_115j · 2024-10-22

**Soundness:** 3
**Presentation:** 2
**Contribution:** 3
**Rating:** 6
**Confidence:** 4

**Summary:**

The authors present a method called Renovo for identifying peptide sequences from mass spectrometry data. The challenge of peptide identification is undeniably complex and significant, and this paper offers a solution that integrates prior sequence knowledge into de novo sequencing. This is achieved by revisiting learned co-occurrences of amino acids. The authors evaluate their method against widely used de novo techniques using standard community benchmarking datasets, demonstrating superior performance.

**Strengths:**

•	The authors advance beyond existing de novo methods by allowing the integration of prior knowledge.

•	They showcase better performance compared to established approaches.

*       The manuscript is well written and easy to follow.

**Weaknesses:**

•	Mixture methods combining de novo and database searches have been well documented (notably Spider 10.1109/CSB.2004.1332434 and BICEPS 10.1074/mcp.M111.014167). This is not a novel contribution of this paper, as such approaches have been recognized for two decades and should be acknowledged as related work.

•	Error-tolerant and open database searches are prevalent in the field but are not discussed here, despite their potential relevance (as noted in the following point).

•	The authors claim that comparing against database searches is not valid, yet they incorporate prior database information into Renovo that is unavailable to other de novo methods, creating an unfair comparison.

**Questions:**

•	The specific database information provided to Renovo during the benchmarks is unclear. What is the size of the database, and to what extent are related sequences excluded (are only identical sequences filtered out, or do filters also apply to those with PTMs, chemical modifications, or single amino acid substitutions)?

•	Can the authors compare their results to an error-tolerant/open database search method, such as MSFragger, which is currently considered the gold standard in the field? It may be beneficial to replicate a typical scenario, such as in a 7/9 species setup, where I could supply a complete database of closely related organisms. For instance, while humans and chickens diverged around 300 million years ago, it has been shown that error-tolerant approaches can effectively bridge that gap. Unless going for very obscure organisms, there are related species available and that should be the starting point.

•	In practical proteomics applications, reliability is crucial, yet the benchmark setup appears limited in this context. Most mass spectra tend to be low quality, and redundancy in measurements is high. Standard practice (and often a requirement for publications in the field) necessitates that peptide identifications exceed a 1% false discovery rate (FDR) threshold. Currently, the authors only provide precision and recall metrics, which do not indicate how many peptides the search algorithm considers reliable. It is not essential for every spectrum to be identified. Can the authors also present the number of peptides identified above the 1% FDR threshold?

---

> ### Author Response · Authors · 2024-11-18
> **Response to Reviewer 115j (1)**
>
> We appreciate your insightful and helpful reviews!
>
>
> > W1: **Mixture methods combining de novo and database searches** have been well documented (notably **Spider** 10.1109/CSB.2004.1332434 and **BICEPS** 10.1074/mcp.M111.014167). This is not a novel contribution of this paper, as such approaches have been recognized for two decades and should be acknowledged as related work.
>
> Thanks for your insightful reviews!
>
> I think we might have different understandings regarding the novelty of ReNovo. As stated in our original text, **"ReNovo is the first-of-its-kind retrieval-based framework for the de novo peptide sequencing task."** In essence, **ReNovo is a de novo peptide sequencing method** that leverages a retrieval-based framework. In contrast, **Spider [1] and BICEPS [2], as you mentioned, are fundamentally database searching methods** that utilize the results from peptide sequencing.
>
> Specifically, we refer to the database used by the ReNovo model as the "datastore" to distinguish it from conventional database search methods. The datastore of ReNovo is constructed using the training data that is employed to train the ReNovo model, which inherently makes it a de novo peptide sequencing method. **ReNovo does not utilize any pre-existing protein database or prior knowledge beyond the training set.** Moreover, ReNovo is capable of generating novel peptides that are not present in the training set, as demonstrated by its performance on the test set (as shown in Table 1 of the original paper), where the training and test sets are completely distinct in terms of peptides.
>
> In contrast, the database search methods you mentioned:
>
> - **SPIDER** [1] is an algorithm that matches sequence tags (obtained through de novo sequencing software) with errors to database sequences for peptide identification. SPIDER’s alignment model requires the use of a database to align a real sequence with a database sequence. As stated in their original work: *"**The protein databases that SPIDER searches are usually large**. For example, the release 42.12 (15-Mar-2004) of the Swiss-Prot database contains 146,193 proteins comprising 53,888,514 amino acids.*" Therefore, **SPIDER fundamentally falls under the category of database search methods**, as it relies on large, externally sourced peptide databases.
>
> - **BICEPS** [2] is a Bayesian Information Criterion-driven error-tolerant peptide database search method designed to overcome species boundaries in peptide identification. In the BICEPS algorithm, sequence tags (generated by de novo methods) are mapped to candidate sequences from a database. Paper [2] showed its performance compared to database search algorithms (such as Mascot) involving a human sample searched against a chicken database. Therefore, **BICEPS remains a database search method, as it relies on an externally sourced database.**
>
> **In conclusion, SPIDER and BICEPS are fundamentally database search methods that rely on external databases, whereas our ReNovo, a de novo peptide sequencing method, belongs to an entirely different paradigm. Therefore, SPIDER and BICEPS do not undermine the novel contribution of ReNovo**.
>
> > W2: **Error-tolerant and open database searches are prevalent in the field but are not discussed here**, despite their ...
>
> As mentioned above, ReNovo is a de novo peptide sequencing method, not a database search method or a variant of it (including error-tolerant and open database searches). Database search methods and de novo peptide sequencing methods are two entirely different paradigms in peptide identification. ReNovo does not utilize any externally sourced protein/peptide database beyond the training dataset, and therefore, it is not suitable for tasks such as error-tolerant or open database searches. **Due to the significant differences in paradigms, we did not discuss works related to error-tolerant or open database searches in the "Related Work" section of the initial version of our paper.**
>
> However, to facilitate readers' understanding, **we have included additional content in the revised paper that discusses error-tolerant/open database search method**. Thank you for your valuable advice!
>
> ---
>
> [1] SPIDER: Software for Protein Identification from Sequence Tags with De Novo Sequencing Error
>
> [2] Overcoming Species Boundaries in Peptide Identification with Bayesian Information Criterion-driven Error-tolerant Peptide Search (BICEPS)
>
> ---
>
> **The next part of our response can be found in the next block.**

---

> ### Author Response · Authors · 2024-11-18
> **Response to Reviewer 115j (2)**
>
> We appreciate your insightful and helpful reviews!
>
> > W3: The authors claim that comparing against database searches is not valid, yet they **incorporate prior database information into Renovo** that is unavailable to other de novo methods, **creating an unfair comparison**.
>
> > Q1: **The specific database information provided to Renovo during the benchmarks is unclear. What is the size of the database,** and...
>
> Another good point!
>
> I think we might have different understandings regarding the concept of ReNovo's datastore. As mentioned in the original paper, ReNovo constructs a datastore that contains information solely derived from the training data. During the inference (de novo peptide sequencing) stage, this datastore is queried. It is crucial to emphasize that **ReNovo fundamentally remains a de novo method, without relying on or incorporating any prior database information**.
>
> Besides, **the comparison in Table 1 is entirely fair, as both ReNovo and the other de novo baseline models utilize information solely from the same training dataset.**
>
> **Regarding the construction process and methodology of ReNovo's datastore (or database),** we have explained it in Section 3.3 of the original paper: After the model training stage, ReNovo utilizes the pre-trained model P(y∣s,p;θ∗) along with the training dataset (S,P,Y) to build the datastore D. The information (i.e., key-value pairs (K,V)) stored in datastore D can be interpreted as "context feature-target amino acid pairs". The "context feature" is a feature vector output by the pre-trained ReNovo model, which encapsulates information from the input MS2 spectra s, precursor p, and the previously predicted peptide sequence y1:t−1. The "target amino acid" y_t represents the ground truth prediction under the corresponding context represented by the context feature.
>
> **Regarding the size of ReNovo's datastore** (or database), it is mentioned in Table 4 of the original paper, as follows:
>
> **Table 4: Datastore statistic.**
> |     | Seven-species Dataset    | Nine-species Dataset | HC-PT Dataset  |
> |-------|:-------:|:-------:|:-------:|
> | Pairs Number | 5,626,944  | 8,456,240 | 3,232,616 |
> | Storage (GB) | 11.16  | 16.77 | 6.42 |
>
> Depending on the scales of datasets, the datastores contain 3.2 ~ 5.6 million pairs and occupied is 6~11 GB storage space. Considering that the datastore is plug-and-play, occupying only storage space rather than memory space, the consumption can be considered negligible.
>
> > Q2: **Can the authors compare their results to an error-tolerant/open database search method,** such as MSFragger, which is ...
>
> What we want to emphasize is that the **database search method and de novo peptide sequencing are two fundamentally different paradigms in peptide identification, each suited for different scenarios. It is unnecessary to conduct a direct performance comparison under identical conditions**. Database searching methods are not suitable for our de novo scenarios, where the peptide sequences in the test set do not exist in any database. Consequently, ReNovo is not appropriate for a direct comparison with database search method, such as MSFragger.
>
> Additionally, none of the baseline models, nor recent works on deep learning-based de novo method [4, 5, 6, 7], have conducted performance comparisons with database searching. I believe that it's both common and reasonable not to compare de novo with database search methods when evaluating the performance of de novo approaches.
>
> ---
>
> [4] De novo peptide sequencing by deep learning (Tran et al., PNAS)
>
> [5] Computationally instrument-resolution independent de novo peptide sequencing for high-resolution devices (Qiao et al., Nature Machine Intelligence)
>
> [6] De Novo Mass Spectrometry Peptide Sequencing with a Transformer Model (Yilmaz et al., ICML 2022)
>
> [7] AdaNovo: Towards Robust \emph{De Novo} Peptide Sequencing in Proteomics against Data Biases (Jun et al., NeurIPS 2024)
>
> ---
>
> **The next part of our response can be found in the next block.**

---

> ### Author Response · Authors · 2024-11-18
> **Response to Reviewer 115j (3)**
>
> We appreciate your insightful and helpful reviews!
>
> > Q3: In practical proteomics applications, **reliability is crucial**, yet the benchmark setup appears limited in this context. ... **Can the authors also present the number of peptides identified above the 1% FDR threshold?**
>
> Thanks for your helpful reviews!
>
> The false discovery rate **(FDR) threshold is a metric widely used in "database search methods,"** where it involves constructing a decoy database from the target database. The FDR threshold represents the extent to which database search methods incorrectly map sequencing results to the decoy database. In contrast, ReNovo falls under the category of "de novo peptide sequencing methods" as it builds a datastore using training data rather than any external data.
>
> Therefore, although reliability is crucial in practical proteomics applications, the FDR threshold, which is widely used in "database search methods," is not applicable to ReNovo (a "de novo peptide sequencing method"). **There has not yet been an well-established method for creating decoy databases and estimating their FDR in de novo** peptide sequencing. Moreover, related works and baselines that also belong to the category of "de novo peptide sequencing methods" [1, 2, 3, 4] have not used the FDR threshold as a metric for evaluating model performance or reliability.
>
> **To better evaluate the reliability of ReNovo, we have added a recall-coverage curve in the revised version of the paper.** This curve offers valuable insights into the performance across varying confidence levels, capturing the model's reliability. You can find this curve in Appendix C of the updated paper.
>
> Additionally, the peptide-AUC metric in original paper is determined by calculating the area under the peptide-level precision-recall curve, providing a comprehensive evaluation. The **peptide-level AUC metric could also offer insights into the model's reliability.**
>
> ---
>
> [1] De novo peptide sequencing by deep learning (Tran et al., PNAS)
>
> [2] Computationally instrument-resolution independent de novo peptide sequencing for high-resolution devices (Qiao et al., Nature Machine Intelligence)
>
> [3] De Novo Mass Spectrometry Peptide Sequencing with a Transformer Model (Yilmaz et al., ICML 2022)
>
> [4] AdaNovo: Towards Robust \emph{De Novo} Peptide Sequencing in Proteomics against Data Biases (Xia et al., NeurIPS 2024)
>
>
> ---
> **Thank you once again for your thorough, insightful, and constructive reviews! If our responses have addressed your concerns fairly, we respectfully hope you might consider raising your score to support our work. If you have any other questions or concerns, we are very willing to engage in further discussion.**

---

> ### Author Response · Authors · 2024-11-25
> **Has our response resolved your concerns?**
>
> Dear Reviewer 115j,
>
> **We appreciate your insightful and helpful reviews, and have made every effort to address the concerns you raised.**
>
> If our response has resolved your concerns and clarified any ambiguities, we respectfully hope you might consider raising the score. Should you have further questions or need additional clarification, we would be happy to discuss them. Thank you again for your time and effort in reviewing our manuscript. Your feedback has been instrumental in improving our research!
>
> Best regards,
>
> Authors

---

> > ### Comment · Reviewer_115j · 2024-11-25
> >
> > Thank you for the very detailled feedback to my questions raised. This is very helpful. At the same time, I do have still some concerns:
> > - as also brought up by eX4t I wonder why no comparison is carried out to database tools. The statement by the authors that this is not possible as db search algorithms can only identify previously known peptides, is just incorrect. Error-tolerant database searches can overcome much larger differences than those of the species in the search here. This should be honestly stated.
> > - related, I am afraid there is still a risk of data leakage in the approach used by the authors. If related peptides are not filtered (and I did not see an answer to that questions) and considering the redundancy of b- and y- ion changes in mass spectra, I will see the exact same information in training and testing e.g. for an N-term modified peptide.
> > - Re FDR: this is a well established statistical metric with usage well beyond proteomics. It is of interest if the risk of having FPs is much higher than the benefits of an additional TP which is the case for the extremely redundant nature of mass spectrometry peptides.
> > Contrary to what the authors states, it is also used for de novo comparisons in the field (e.g. https://www.mcponline.org/article/S1535-9476(24)00139-7/fulltext for a comprehensive framework).
> > For these reasons, I ask for your understanding that I am currently not adjusting my score.

---

> ### Author Response · Authors · 2024-11-30
> **Response to Reviewer 115j (1)**
>
> Thank you for the valuable feedback. I will address each of your concerns in detail.
>
> > ... **I wonder why no comparison is carried out to database tools.**
>
> Thanks for your to-the-point reviews!
>
> **What we want to emphasize is that database search methods and de novo methods represent two fundamentally different paradigms in peptide identification, each suited for specific scenarios.** Mainstream database search methods are primarily employed to identify known peptides, relying on large-scale, external databases. In contrast, de novo method is capable of identifying novel peptides without depending on large-scale external databases. This approach is particularly valuable in situations where peptide databases are unavailable, such as in the sequencing of antibodies and novel antigens, as well as in the sequencing of metaproteomes lacking established databases.
>
> **In other words, since database search methods and de novo methods are suited for entirely different scenarios, it is unnecessary to conduct a direct performance comparison under identical conditions.**
>
> Additionally, none of the de novo models mentioned in our paper (deepnovo, pointnovo, casanovo, helixnovo, adanovo) have performed performance comparisons with database search methods. **I believe that it is both common and reasonable not to directly compare de novo methods with database search when evaluating the performance.**
>
> > The statement by the authors that this is not possible as db search algorithms can only identify previously known peptides, is just incorrect. **Error-tolerant database searches can overcome much larger differences than those of the species in the search here**. This should be honestly stated.
>
> **We fully acknowledge the correction you have pointed out**. Despite representative database search methods are generally limited to identifying known peptides. Nevertheless, there are approaches such as error-tolerant database searches, which can overcome much larger differences than those of the species in the search here. **To facilitate readers' understanding, we have included additional content in the revised paper that discusses error-tolerant/open database search method.**
>
> However, it is important to note that **error-tolerant database searches still fall within the category of database search methods and remain dependent on large-scale, external databases to achieve optimal results**. For instance, the database search methods you mentioned:
>
> - SPIDER : As stated in their original work: "*The protein databases that SPIDER searches are usually large. For example, the release 42.12 (15-Mar-2004) of the Swiss-Prot database contains 146,193 proteins comprising 53,888,514 amino acids.*" Therefore, SPIDER relies on large, externally sourced peptide databases.
>
> - MSFragger : Notably, *MSFragger utilizes the human UniProtKB database, which, as of Release 2024_05 of 02-Oct-2024, comprises 248,266,673 sequence entries encompassing 88,056,623,818 amino acids*.
>
> In contrast, ReNovo does not rely on any external, large-scale protein databases or prior knowledge beyond its training set. Furthermore, ReNovo is capable of generating novel peptides that are not part of the training data. **This distinction indicates that the scenarios for employing database search methods and de novo methods are fundamentally different**.
>
> ---
>
> **The next part of our response can be found in the next block.**

---

### Official Review · Reviewer_Hu9o · 2024-10-31

**Soundness:** 3
**Presentation:** 3
**Contribution:** 3
**Rating:** 6
**Confidence:** 4

**Summary:**

This paper Introduces a novel approach for peptide identification in proteomics, where conventional methods rely heavily on pre-constructed peptide databases, limiting their scope when dealing with novel peptides. ReNovo, leveraging a retrieval-based framework, bypasses this limitation by building a datastore from training data and retrieving relevant information during inference. This approach combines the strengths of both database search and de novo sequencing, enabling it to predict peptides absent from existing databases.

**Strengths:**

The paper presents strong motivation for combining database search with de novo sequencing, marking the first integration of the Retrieval-Augmented Generation (RAG) method into protein sequence prediction, which adds a novel dimension to the field. The authors effectively demonstrate the proposed method's success, showcasing the potential of integrating RAG into bioinformatics, particularly in areas with extensive databases.

**Weaknesses:**

1. The paper primarily employs a token-level RAG method from NLP but does not thoroughly discuss its origins or previous related work. Specifically, there is no comprehensive section on RAG methods, with most background instead focused on de novo methods. The approach closely resembles the KNN-LM paper, which originally proposed storing context and next tokens in a datastore; however, this foundational work is not referenced.

Additionally, the paper omits discussion on recent advancements in de novo methods, such as GraphNovo, PepNet, PrimeNovo, and InstaNovo, all of which are documented in Bittremieux et al.'s survey on de novo peptide sequencing methods, if author needs to refer to a good source.

2. Missing Performance Comparisons: The model performance comparisons in the main table do not adequately reflect recent advancements in the field, with several relevant methods from the past year omitted. Including a broader range of models would provide a clearer view of the current state of the art.

3. Clarification is needed on the training and testing datasets: whether a single model was trained on a single dataset and used for all number reporting or if separate models were trained for each dataset? Additionally, it would be helpful to confirm if the reported performance numbers for baseliens are replications or directly quoted from original papers. For instance, discrepancies are noted between some results here and those reported for CasaNovo paper i read from their published journal.

4. For the case study, a direct alignment between the predicted and retrieved spectra would make the comparison clearer. The current presentation lacks an easily interpretable visual comparison of their similarities.

5. If the first stage of training essentially reproduces the Casanovo model, the authors could consider using available open-source models for this step. Testing stronger baseline models could indicate whether enhanced context representations lead to better context features in the datastore. No need to re-train casanovo as first stage IMO.

6. Including a recall-coverage curve could offer valuable insights into performance across varying confidence levels, which is especially useful for biologists relying on model certainty for predictions on unlabeled data. This would provide more depth than single averaged metrics, which may not fully capture the model's reliability.

**Questions:**

The authors use training data to construct the datastore, but I believe this approach could benefit from further discussion and refinement. In mainstream RAG-based methods, the datastore is generally maintained as an independent entity, distinct from the training data for language models. This separation allows the datastore to act as an “additional” resource that complements a model’s learned knowledge. When the same dataset is used for both training the language model and constructing the datastore, much of the useful information may already be encoded in the model weights through the training phase, reducing the unique value of “retrieval.” Specifically, the language model’s objective—next-token prediction—already captures “context-target” relationships by predicting the next token based on the previous context. Token-level retrieval, in this context, could become redundant, as the model effectively performs this retrieval autonomously. I strongly recommend the authors separate training data from datastore data to better evaluate the impact of retrieval on de novo sequencing. It would also be useful to see how a larger datastore, too complex to incorporate into training, might act as an external resource that enhances model performance.



During inference, could you clarify how the top K “next target” retrievals assist the model? Formulas 9-10 lack clarity in this respect. Simplifying the formulas and using clearer explanations would improve comprehension, as sophisticated formulas are only effective if they are accompanied by a clear rationale. Specifically, does the probability for each amino acid derive directly from the retrieved targets in the output layer, or is it calculated after embedding these K tokens into the input layers?

---

> ### Author Response · Authors · 2024-11-23
> **Response to Reviewer Hu9o (1)**
>
> We greatly appreciate your insightful comments. I will address each of your concerns in detail.
>
> > W1: The paper primarily employs a token-level RAG method from NLP but does not thoroughly discuss its origins or previous related work. Specifically, there is no comprehensive section on RAG methods, ...
>
> Thanks for your insightful reviews!
>
> It must be acknowledged that RAG has demonstrated SOTA performance in numerous generative tasks. The ReNovo model mentioned in this paper draws on an important concept from RAG: enhancing the model's generative capabilities by explicitly retrieving existing information during inference rather than relying solely on the knowledge implicitly stored in the model's parameters.
>
> However, I believe that the proposed method, **ReNovo, is novel compared to recent advances in RAG. The retrieval-based inference in ReNovo differs significantly from RAG models in the following ways:**
>
> - In LLMs, RAG typically involves dynamically integrating up-to-date external information from across the web, effectively incorporating external data. In contrast, **the information retrieved by ReNovo originates solely from training data, rather than from any external, domain-specific, or test dataset-related sources.** This approach serves two purposes: first, to ensure a fair comparison with other de novo baselines, and second, to simulate real-world de novo applications—namely, directly interpreting mass spectra to infer peptides independently of any external peptide database. This independence is crucial and adheres to de novo scenarios where peptide databases are unavailable.
>
> - **Our downstream task focuses on de novo peptide sequencing, which is fundamentally different from the typical tasks of RAG** (such as question answering or dialogue response generation), given the distinct data formats involved—mass spectra versus text/images. In LLMs, RAG methods often use pre-trained encoders to process input information, but this approach cannot be directly applied to MS2 data. Therefore, the way ReNovo processes information (see Section 3.1 and Section 3.2) is significantly different from how RAG.
>
> **I believe that the proposed ReNovo method is novel compared to recent advances in RAG**, and the RAG methods used in LLMs cannot be directly applied to de novo peptide sequencing, as no applicable database exists. Given these significant differences, we did not mention RAG methods in the original paper.
>
> To aid readers in comprehending our works and further clarify the novelty of our paper, **we have added a section discussing related RAG research in the revised version of the paper**. This section is now available in Section 2.3 of the updated manuscript.
>
> > W1: Additionally, the paper omits discussion on recent advancements in de novo methods, such as GraphNovo, PepNet, PrimeNovo, and InstaNovo, ...
>
> > W2: Missing Performance Comparisons: The model performance comparisons in the main table do not adequately reflect recent advancements in the field, ...
>
> Thanks for your to-the-point reviews!
>
> Firstly, **our baseline model includes the most recent models such as HelixNovo [1] (published at BIB, April 2024), CasaNovo (published in Nature Communications, July 2024), and AdaNovo [2] (published at NeurIPS, October 2024)**, both published this year (2024) and claim SOTA performance in their respective papers, providing significant reference value. These models are more recent compared to those like GraphNovo (2023), PepNet (2023), and InstaNovo (2023).
>
> Secondly, **reproducing some of the models you mentioned is quite challenging**. For instance, training GraphNovo involves graph construction, which requires an enormous amount of memory. As noted on GraphNovo's official GitHub, "*In case you plan to build a graph from scratch, ensure that you have access to a server with over 500GB of RAM.*" This memory demand is astonishingly high compared to our baseline models and exceeds the limits of most servers, including our own device. The InstaNovo model, on the other hand, has a large number of parameters, making it prone to overfitting when trained directly on our experimental dataset, leading to inferior results that are not suitable for performance comparison. Additionally, as far as we know, PrimeNovo has neither made its implementation code open-source nor released its checkpoints, so we are currently unable to reproduce this model.
>
> If, in the future, the authors of these papers release or refine their implementation codes, making a direct performance comparison between ReNovo and these models feasible, we will indeed include these models. Thank you very much for your suggestions!
>
> ---
>
> [1] Introducing π-helixnovo for practical large-scale de novo peptide sequencing. (Yang et al., Briefings in Bioinformatics 2024)
>
> [2] AdaNovo: Towards Robust \emph{De Novo} Peptide Sequencing in Proteomics against Data Biases (Jun et al., NeurIPS 2024)
>
> ---
>
> **The next part of our response can be found in the next block.**

---

> ### Author Response · Authors · 2024-11-23
> **Response to Reviewer Hu9o (2)**
>
> We greatly appreciate your insightful comments. I will address each of your concerns in detail.
>
> > W3: Clarification is needed on the training and testing datasets: whether a single model was trained on a single dataset and used for all number reporting or if separate models were trained for each dataset?
>
> Thanks for your helpful reviews!
>
> To be precise, **separate models were trained for each dataset.** We used three datasets: the Seven-species Dataset, the Nine-species Dataset, and the HC-PT Dataset. Each dataset was divided into training, validation, and testing sets. Especially, in the case of the Seven-species Dataset, all models (including ReNovo) were trained on the training set of the Seven-species Dataset and subsequently tested on the testing set of the Seven-species Dataset. The same approach was applied to the other two datasets.
>
> > W3: Additionally, ..., discrepancies are noted between some results here and those reported for CasaNovo paper i read from their published journal.
>
> **I believe these discrepancies can be attributed to differences in the datasets**. In the case of the Nine-species Dataset used in both ReNovo and CasaNovo [1], the versions is different: the Nine-species Dataset used by ReNovo (identifier: MSV000081382), while the Nine-species Dataset used by CasaNovo (identifier: MSV000090982) differs. More importantly, the Nine-species Dataset in the CasaNovo paper [1] introduces a data leakage issue. Specifically, the ICML version of Casanovo [4] states in Section 4.2: "*among the approximately 26,000 unique peptide labels associated with the spectra in the test set, there is a 7% overlap with around 250,000 unique peptide labels associated with the training spectra*". In other words, **our evaluation datasets are more challenging because they do not have any data leakage**. The presence of data leakage also contradicts the de novo application scenario, where the peptide to be predicted is absent from existing databases.
>
> Different datasets and varying versions of the same dataset can hinder fair comparisons, making it difficult to assess the performance of different de novo methods. In our study, the reported performance numbers in Table 5 in our original paper are based on unified datasets and partitioning strategies, allowing for a more accurate evaluation of the models' true performance.
>
> **Furthermore, the datasets we utilized (Seven-species Dataset, Nine-species Dataset, and HC-PT Dataset) are more comprehensive compared to the dataset used in the CasaNovo paper [1]** (Nine-species Dataset). This comprehensiveness contributes to a more robust performance assessment and enables a clearer understanding of each method's capabilities.
>
> > W4: For the case study, a direct alignment between the predicted and retrieved spectra would make the comparison clearer. The current presentation lacks an easily interpretable visual comparison of their similarities.
>
> Thanks for your insightful reviews!
>
> Although the ReNovo model is a retrieval-based framework, it does not directly compute the alignment between the predicted spectra and the retrieved spectra. Instead, the model employs the context feature $f({y}_{1:t-1},\mathbf{s},\mathbf{p};\theta^{*})$, generated by the Peptide Decoder, to perform retrieval. As indicated by the expression, the context feature is a feature vector that encapsulates information from the input MS2 spectra s, the precursor p, and the previously predicted peptide sequence y1:t−1.
>
> To provide a easily interpretable visual comparison of the similarities between the retrieved context features and those generated by ReNovo for the case study in Section 4.6, we applied t-SNE to project the context features onto a two-dimensional plane. **The updated visualization results are included in the revised version of the paper, located in Appendix D and Figure 7**.
>
> As highlighted in the case study described in Table 5 of the original paper, without retrieval, the model incorrectly predicts the amino acid ‘E’. However, with retrieval, the ReNovo model retrieves two possible target amino acids: ‘D’ and ‘E’, represented by blue and orange color in the Figure 7, respectively. Notably, the cluster corresponding to amino acid ‘D’ is larger and carries greater weight. By applying the weighting calculations outlined in Equations 9 and 10, the model assigns a higher weight to amino acid ‘D’, resulting in a final prediction of ‘D’ instead of ‘E’. This demonstrates how ReNovo effectively utilizes datastore information to accurately predict the entire peptide sequence.
>
> ---
>
> [1] De Novo Mass Spectrometry Peptide Sequencing with a Transformer Model (Yilmaz et al., ICML 2022)
>
> [2] De novo peptide sequencing by deep learning (Tran et al., PNAS)
>
> ---
>
> **The next part of our response can be found in the next block.**

---

> ### Author Response · Authors · 2024-11-23
> **Response to Reviewer Hu9o (3)**
>
> We greatly appreciate your insightful comments. I will address each of your concerns in detail.
>
> > W6: Including a recall-coverage curve could offer valuable insights into performance across varying confidence levels, ... This would provide more depth than single averaged metrics, which may not fully capture the model's reliability.
>
> Thanks for your helpful reviews!
>
> To better evaluate the reliability of ReNovo, **we have added a recall-coverage curve in the revised version of the paper. You can find this curve in Appendix C of the updated paper.** This curve offers valuable insights into the performance across varying confidence levels, capturing the model's reliability.
>
> > Q1: The authors use training data to construct the datastore, .... **When the same dataset is used for both training the language model and constructing the datastore, much of the useful information may already be encoded in the model weights through the training phase, reducing the unique value of “retrieval.”** ... **I strongly recommend the authors separate training data from datastore data to better evaluate the impact of retrieval on de novo sequencing.** It would also be useful to see how a larger datastore, too complex to incorporate into training, might act as an external resource that enhances model performance.
>
> Thanks for your insightful reviews!
>
> The ReNovo model mentioned in this paper draws on important concepts from RAG: (1) Knowledge does not need to be stored implicitly within model parameters; instead, it is explicitly stored in the datastore and can be retrieved during the inference phase in a plug-and-play fashion, enhancing the model's generative capabilities. (2) By generating sequence based on retrieved references rather than from scratch, the ReNovo model reduces the inherent difficulty of generating coherent and contextually accurate sequence.
>
> Therefore, **even though the data retrieved by ReNovo during inference has already appeared in its training data and may have been encoded in the model's weights, leveraging information retrieved from the datastore during inference still proves effective** in reducing the complexity of the generation task and enhancing the model's generative capacity. As a result, the datastore constructed from the training data remains highly valuable for the ReNovo model. **This is further validated by the ablation experiments presented in Section 4.4 of the original paper.**
>
> **To align with your suggestion and further investigate the effect of datastore size on ReNovo's performance, we conducted additional experiments using a larger dataset** (beyond the training set). Specifically, we utilized a larger dataset from [1] that includes the same species, excluding data belonging to the same species as the test set, to construct the datastore. The training set, test set, and other configurations were kept unchanged. The sizes of the two datastores and the corresponding ReNovo performance metrics are detailed in the following **Table Re1**:
>
> **Table Re1: The performance of the ReNovo model across various datastore sizes.**
> | Datastore Pairs | Datastore Size | Peptide Precision | Peptide AUC | AA Precision | AA Recall |
> |:-------:|:-------:|:-------:|:-------:|:-------:|:-------:|
> | 8,456,240 | 16.77 GB  | 0.568 | 0.528 | 0.770 | 0.769  |
> | 22,646,374 | 44.89 GB| 0.696 | 0.656 | 0.836 | 0.834 |
>
> It is evident that leveraging a larger and more comprehensive datastore, rather than limiting it to the training set, has led to a significant improvement in ReNovo's performance. Thanks for your helpful and constructive suggestions!
>
> ---
>
> [1] De novo peptide sequencing by deep learning (Tran et al., PNAS)
>
> ---
>
> **The next part of our response can be found in the next block.**

---

> ### Author Response · Authors · 2024-11-23
> **Response to Reviewer Hu9o (4)**
>
> We greatly appreciate your insightful comments. I will address each of your concerns in detail.
>
> > Q2: During inference, could you clarify how the top K “next target” retrievals assist the model? Formulas 9-10 lack clarity in this respect. .... Specifically, does the probability for each amino acid derive directly from the retrieved targets in the output layer, or is it calculated after embedding these K tokens into the input layers?
>
> Another good point!
>
> Next, I will provide a detailed explanation of the meaning behind Formulas 9 and 10. The intuition behind Formulas 9 to 10 is to convert the retrieved context feature - target amino acid pairs into a distribution over the amino acid vocabulary, then aggregating over multiple occurrences of the same vocabulary item.
>
> During the inference phase, ReNovo's Peptide Decoder outputs the context feature $f(y1:t−1,s,p;θ∗)$, which is a high-dimensional vector. This context feature is then employed to query the datastore D to identify the K nearest neighbors $N={(k_j,v_j),j∈(1,2,…,K)}$, where $k_j$ and $v_j$ represent the retrieved context feature and the corresponding target amino acid, respectively.
>
> We use the Euclidean distance $d(k_j,f(y1:t−1,s,p;θ∗))$ to quantify the similarity between the retrieved context feature $k_j$ and the context feature $f(y1:t−1,s,p;θ∗)$ output by the ReNovo. A smaller Euclidean distance $d(·,·)$ indicates that the retrieved feature is closer to the ReNovo's output feature and, therefore, more relevant, warranting a higher weight. Specifically, we adopt a common technique in classification tasks known as temperature scaling softmax [1], where we use $exp⁡(−d/T)$ to transform the retrieved distance $d(k_j,f(y1:t−1,s,p;θ∗))$ into a weight, with $T$ being the hytperparameter.
>
> Since we have retrieved K nearest neighbors $N={(k_j,v_j),j∈(1,2,…,K)}$, it is necessary to compute K values of $exp⁡(−d/T)$ and aggregate those corresponding to the same retrieved target amino acid $v_j$.
>
> Specifically, for amino acid 'G', the weight in ReNovo can be expressed as:
>
> $$
> p_{\text{kNN}}( y_t = G \mid {y}_{1:t-1},{s},{p};\theta^{*})
> $$
>
> $$
> = \sum_{(k_j, v_j) \in {N}} {1}_{v_j = G} \exp ( \frac{-d(k_j,  f( y1:t-1 ,s,p;θ∗)  )}{T} )
> $$
>
> The indicator function $1_{v_j=G}$ specifies that only pairs from the K nearest neighbors N satisfying $v_j=G$ are considered. If we extend this to include all amino acids, meaning $y_t∈AA$, where $AA$ denotes the whole amino acid vocabulary, the above formula can be generalized as:
>
> $$
> p_{\text{kNN}}(y_t \mid {y}_{1:t-1},{s},{p};\theta^{*})
> $$
>
> $$
> = \sum_{(k_j, v_j) \in {N}} {1}_{y_t = v_j} \exp ( \frac{-d(k_j,  f( y1:t-1 ,s,p;θ∗)  )}{T} ), y_t∈AA
> $$
>
> The above formula corresponds to Formula 9. Since ReNovo's generation at step t essentially represents a classification task (classifying the t-th amino acid), it is necessary to produce a probability distribution. This requires an additional normalization step, as represented by Formula 10:
>
> $$
> p_{\text{aa}}\left( y_t \mid {y}_{1:t-1},{s},{p};\theta^{*} \right)
> $$
>
> $$
> = \frac{p_{\text{kNN}}( y_t \mid y1:t-1 ,s,p;θ∗ )}{\sum_{y_j \in {AA}} p_{\text{kNN}}( y_j \mid y1:t-1 ,s,p;θ∗ )}, y_t \in {AA}
> $$
>
> We provides a detailed explanation of Formulas 9 and 10. Formula 9 outlines the calculation of weights for each amino acid by aggregating the contributions of retrieved neighbors based on their distance to the decoder's output. Formula 10 normalizes these weights to form a valid probability distribution over the amino acid vocabulary, facilitating the classification process for the t-th amino acid during ReNovo's inference stage.
>
> ---
>
> [1] On calibration of modern neural networks. (Guo et al., ICML 2017)
>
> ---
>
> **Thank you once again for your thorough, insightful, and constructive reviews! If our responses have addressed your concerns, we respectfully hope you might consider raising your score to support our work.**

---

> ### Author Response · Authors · 2024-11-25
> **Has our response resolved your concerns?**
>
> Dear Reviewer Hu9o,
>
> **We appreciate your insightful and helpful reviews, and have made every effort to address the concerns you raised.**
>
> If our response has resolved your concerns and clarified any ambiguities, we respectfully hope you might consider raising the score. Should you have further questions or need additional clarification, we would be happy to discuss them. Thank you again for your time and effort in reviewing our manuscript. Your feedback has been instrumental in improving our research!
>
> Best, Authors

---

> > ### Comment · Reviewer_Hu9o · 2024-11-26
> > **To authors:**
> >
> > Thank you for your detailed response. I have reviewed it thoroughly and would like to share some additional questions and concerns:
> >
> > >  we believe that the proposed ReNovo method is novel compared to recent advances in RAG, and the RAG methods used in LLMs cannot be directly applied to de novo peptide sequencing, as no applicable database exists. Given these significant differences, we did not mention RAG methods in the original paper.
> >
> > I do not question the novelty of this method. However, acknowledging where the work derives some of its ideas can help readers better understand your unique contributions. I strongly suggest adding a section to discuss related RAG work in the NLP field, specifically KNN-LM.
> >
> >
> > >  reproducing some of the models you mentioned is quite challenging.
> >
> > Correct me if I’m wrong, but I believe most de novo work uses the same benchmark datasets, which means the results can often be quoted?
> > Typically, including some performance with a footnote indicating that the numbers are quoted, and noting the difficulty of reproducing the results, is common practice in the ML community. I encourage you to consider this approach.
> >
> > Your responses to my other questions adequately addressed my concerns. I also find it quite surprising that adding an additional data in retrieval system resulted in such a large performance gain (over 10%?). This finding is of interest for the research community, and I suggest emphasizing it and including it in the main sections of the paper, along with a detailed discussion. A revised manuscript would be greatly appreciated.
> >
> > I have updated my scores, but I welcome further discussion on the concerns I have outlined here.

---

### Official Review · Reviewer_xfGv · 2024-11-02

**Soundness:** 2
**Presentation:** 2
**Contribution:** 3
**Rating:** 8
**Confidence:** 4

**Summary:**

The paper introduces a new method for predicting peptide sequences from MS/MS spectra, with a key innovation in combining two complementary paradigms in computational proteomics: database search and de novo sequence generation. Specifically, the paper introduces the concept of a datastore, which can be used in conjunction with an autoregressive transformer model. After model training, the datastore stores outputs from the trained model across the entire training dataset, which are then used during test-time inference via k-NN indexing. The method empirically outperforms several state-of-the-art approaches across three datasets, requiring only minor additional storage and time complexity. The authors include ablation studies on key hyperparameters. The code is available in the Supplementary materials.

**Strengths:**

Originality

The work’s originality mainly lies in the combination of two primary paradigms commonly applied to peptide MS/MS spectra annotation: database retrieval and de novo sequence generation.

Quality

The paper is overal of a good quality, particularly in terms of dataset preparation, benchmarking the proposed method against prior approaches, and the analysis of results.

Clarity

The paper is well-structured, clearly defines the motivation for the proposed methods, and provides a thorough analysis of the results.

Significance

The work offers a new perspective on the problem of de novo peptide sequencing, which could have a strong impact on future research in this area.

**Weaknesses:**

- The main concern with the proposed approach is whether the state-of-the-art results are not a consequence of the model’s higher capacity rather than its design. To address this, it would be relevant to see, for example, results from the following experiment: evaluate the ReNovo without Retrieval model of larger size (e.g., by increasing the transformer’s number of layers and hidden dimensionality) and compare it to ReNovo, as illustrated, for instance, in Figure 3.

- Another related concern is the train-test splitting strategy. The text states that “peptides appearing in the testing set are completely distinct from those in the training set.” It would be valuable to assess the method’s performance under a stricter splitting criterion, such as one based on a sequence similarity threshold rather than sequence identity. Such an experiment would demonstrate whether the improved performance is due to generalization rather than overfitting to the training dataset through the “datastore” approach. This could be also tested by measuring the model’s improvement on test sequences that lack similar sequences in the training set.

- Certain essential parts of the text lack clarity.
    - The term “context features” is central to the work but is not clearly defined. Does this refer to an embedding from the last transformer decoder layer? What does $f$ in line 213 signify?
    - The “ReNovo with Residual” model is not fully defined. How exactly are “ReNovo without Retrieval” and “ReNovo” configured?

**Questions:**

- How was the case study example in Table 5 selected? Is it possible to quantify the number of similar improvements across the entire test set?
- What were the hyperparameters used for training, specifically the number of epochs and learning rate?

---

> ### Author Response · Authors · 2024-11-17
> **Response to Reviewer xfGv (1)**
>
> We greatly appreciate your insightful comments. I will address each of your concerns in detail.
>
> > Q1: The main concern with the proposed approach is **whether the state-of-the-art results are not a consequence of the model’s higher capacity rather than its design**. To address this, ...
>
> Thanks for your insightful reviews!
>
> To address your concerns and explore the impact of model size on performance, we conducted experiments on the Nine-species Dataset and compared our model against HelixNovo and AdaNovo(SOTA baseline). By varying the transformer's number of layers and hidden dimensionality in Renovo, **we adjusted the parameter count and assessed their performance**, as depicted in **Table Re1**.
>
>
> **Table Re1: Parameters and Performance of Different Models.**
> | Model Name   | Parameter Size   | Peptide Precision   | Peptide AUC |
> |-------|:-------:|:-------:|:-------:|
> | ReNovo_A | 47.3 M | 0.559 | 0.520 |
> | ReNovo_B | 68.0 M | 0.582 | 0.548 |
> | ReNovo_C | 92.2 M | 0.561 | 0.524 |
> | HelixNovo | 47.3 M | 0.517 | 0.453 |
> | AdaNovo | 66.3 M | 0.505 | 0.469 |
>
> From **Table Re1**, it is evident that the ReNovo model significantly outperforms baseline models with similar parameter counts (ReNovo_A vs. HelixNovo, ReNovo_B vs. AdaNovo). On the other hand, a comparison of the performances of ReNovo_A, ReNovo_B, and ReNovo_C reveals that merely increasing the parameter count does not notably enhance ReNovo's performance. This is likely due to potential overfitting issues with overly large models, a challenge that may be more pronounced under the leave-one-out approach used in our original experiments.
>
> I believe this **demonstrates that the SOTA results achieved by ReNovo are attributable to its retrieval-based design rather than the size of its parameters**.
>
>
> > Q2: Another related concern is the train-test splitting strategy ... **It would be valuable to assess the method’s performance under a stricter splitting criterion, such as one based on a sequence similarity threshold**  ....
>
> Another good point!
>
> To address your concerns and explore the model's performance on test sequences that lack similar sequences in the training set, we conducted experiments on the Nine-species Dataset, using AdaNovo (the best baseline for this dataset) and CasaNovo for comparison. We began by calculating the Levenshtein distance (a widely adopted method for measuring the similarity between two strings) for each test sequence against all sequences in the training set. We then identified the minimum Levenshtein distance for each test sequence and filtered out test sequences with a minimum Levenshtein distance less than or equal to 3 and 5, respectively, resulting in **Nine-species Test Dataset (>3) and Nine-species Test Dataset (>5). These datasets, having removed potential peptide-level data leakage**, serve to validate "the model’s improvement on test sequences that lack similar sequences in the training set." The results are presented in **Table Re2**.
>
> **Table Re2: Performance Comparison on Filtered Test Datasets.**
> | Model Name   | Test Set  | Peptide Precision   | Peptide AUC | AA Precision   | AA Recall |
> |-------|-------|:-------:|:-------:|:-------:|:-------:|
> | CasaNovo  | Nine-species Test Dataset (>3) | 0.369 | 0.324 | 0.638 | 0.637|
> | AdaNovo | Nine-species Test Dataset (>3) | 0.398 | 0.356 | 0.648 | 0.656 |
> | **ReNovo** | Nine-species Test Dataset (>3) | **0.465** | **0.423** | **0.721** | **0.720** |
> | CasaNovo  | Nine-species Test Dataset (>5) | 0.302 | 0.254 | 0.599 | 0.598 |
> | AdaNovo  | Nine-species Test Dataset (>5) | 0.333 | 0.292 | 0.613 | 0.618 |
> | **ReNovo** | Nine-species Test Dataset (>5) | **0.396** | **0.351** | **0.688** | **0.686** |
>
> From Table Re2, we observe that the performance of all models declines on the filtered test datasets, which is expected as the test set contains fewer similar data to the training set. However, it is also evident that ReNovo still outperforms the SOTA baseline models significantly on the same filtered test sets. This clearly **indicates that the improved performance of ReNovo is due to generalization rather than overfitting to the training dataset through the "datastore" approach**.
>
> On the other hand, the **"leave-one-out data split by species" setting is widely used** in nearly all de novo sequencing methods [1,2,3,4], and we adhere to the same settings. Furthermore, the leave-one-out approach simulates real-world applications of de novo models: training on known species and sequencing on unannotated species.
>
> ---
>
> [1] De novo peptide sequencing by deep learning (Tran et al., PNAS)
>
> [2] Computationally instrument-resolution independent de novo peptide sequencing for high-resolution devices (Qiao et al., Nature Machine Intelligence)
>
> [3] De Novo Mass Spectrometry Peptide Sequencing with a Transformer Model (Yilmaz et al., ICML 2022)
>
> [4] AdaNovo: Towards Robust De Novo Peptide Sequencing in Proteomics against Data Biases (Xia et al., NeurIPS 2024)

---

> ### Author Response · Authors · 2024-11-17
> **Response to Reviewer xfGv (2)**
>
> We greatly appreciate your insightful comments. I will address each of your concerns in detail.
>
> > Q3: **Certain essential parts of the text lack clarity**: The term **“context features”** is central to the work but is not clearly defined. Does this refer to an embedding from the last transformer decoder layer? What does **f in line 213** signify? The **“ReNovo with Residual”** model is not fully defined. How exactly are **“ReNovo without Retrieval”** and **“ReNovo”** configured?
>
> Thanks for your to-the-point reviews!
>
> **About context features and f:** As we described in Section 3.3 (Lines 211-215) of the original paper, context features are derived from the output of ReNovo's Peptide Decoder and can be represented as f(y1:t−1, s, p; θ ∗ ). This notation refers to a high-dimensional feature vector that encapsulates information from the input MS2 spectra s, precursor p, and the previously predicted peptide sequence y1:t−1.  The **understanding that "this refer to an embedding from the last transformer decoder layer" is indeed correct**. In this context, **f can be viewed as the ReNovo model with θ ∗ as the trainable parameters**. The output to f is the context features.
>
> **About "ReNovo with Residual", "ReNovo without Retrieval" and "ReNovo":** As we explained in Section 4.4 (Lines 364-374) of the original paper: **(1) "ReNovo w/o Retrieval"** refers to the trained ReNovo model that utilizes only the logits from the Peptide Decoder without relying on datastore retrieval. In this case, the model accomplishes the task without engaging in datastore building stage and retrieval-based inference stage. **(2) "ReNovo"** indicates that the ReNovo model utilizes the trained parameters to construct a datastore. During inference, the ReNovo model retrieves from the datastore to make the final prediction, which can be explicitly represented by Equations 9 and 10 in the original paper. This aspect is a core contribution of our work. **(3) "ReNovo with Residual"** denotes an approach where, based on "ReNovo", both the logits from the Peptide Decoder and the datastore retrieval are used for inference. We simply sum the logits from these two output, similar to a common residual connection technique in deep learning architectures.
>
> **In summary, the differences lie solely in the inference methods used:** "ReNovo without Retrieval" relies on the logits from the Peptide Decoder alone during inference. "ReNovo" incorporates the retrieved information from the datastore to make the final prediction. "ReNovo with Residual" combines both approaches by simply adding the logits from them. We introduced these concepts primarily to conduct ablation studies that validate the contribution of each component of ReNovo. Our main focus is on "ReNovo" itself. The other terms serve to clarify different configurations of the model but are not the central emphasis of the paper.
>
> > Q4: How was the case study example in Table 5 selected? Is it possible to quantify the number of similar improvements across the entire test set?
>
> **The case study example in Table 5 is a random selection** from all samples that align with the idea that "ReNovo can make more accurate predictions with the assistance of the datastore." It aims to illustrate that by leveraging an established datastore, the accuracy of peptide predictions can be significantly enhanced.
>
> Therefore, to "quantify the number of similar improvements across the entire test set," I believe that the ablation study presented in the original text (Section 4.4 Lines 364-374) is sufficiently solid. This study investigates the performance differences between the two variants, "ReNovo" and "ReNovo without Retrieval (ReNovo w/o Retrieval)". Consequently, **the performance improvement of "ReNovo" compared to "ReNovo w/o Retrieval" can be interpreted as a quantification of the number of similar enhancements** across the entire test set. Specifically, we have outlined the performance differences in Table Re3 below and in Figure 3 of the original paper.
>
> **Table Re3: Performance Comparison on Filtered Test Datasets.**
> | Model Name   | Test Set | Peptide Precision | AA-level Precision |
> |-------|-------|-------|-------|
> | ReNovo without Retrieval | Seven-species Dataset | 0.176| 0.425 |
> | ReNovo | Seven-species Dataset | 0.272 | 0.510 |
> | ReNovo without Retrieval | Nine-species Dataset | 0.462 | 0.695|
> | ReNovo | Nine-species Dataset | 0.568 | 0.770 |
> | ReNovo without Retrieval | HC-PT Dataset | 0.381| 0.554 |
> | ReNovo | HC-PT Dataset | 0.467 | 0.651 |
>
> **The second part of our response can be found in the next block.**

---

> ### Author Response · Authors · 2024-11-17
> **Response to Reviewer xfGv (3)**
>
> We greatly appreciate your insightful comments. I will address each of your concerns in detail.
>
> > Q5: **What were the hyperparameters** used for training, specifically the number of epochs and learning rate?
>
> Another good point!
>
> In terms of specific hyperparameters:
> - We set the maximum number of epochs to 50 and selected the model corresponding to the epoch with the lowest validation loss for testing.
> - We applied a learning rate scheduler that utilizes a linear warm-up followed by cosine decay. Specifically, the learning rate was linearly increased from zero to 0.0005 and then decayed to zero following a cosine function.
> - To mitigate overfitting on smaller datasets, we incorporated dropout [2] (ranging from 0.03 to 0.05) and L2 regularization (weight decay) [3], given the varying sizes of the three datasets.
> - We also employed a technique commonly used in classification tasks called training label smoothing [1]. This technique applies slight smoothing to the training labels, enhancing the model's generalization ability.
> - During training, the batch size was set to 32.
>
> ---
>
> [1] Rethinking the inception architecture for computer vision (Szegedy et al., ECCV 2016)
>
> [2] Dropout: a simple way to prevent neural networks from overfitting (Srivastava et al., JMLR 2014)
>
> [3] A simple weight decay can improve generalization (A Krogh et al., NeurIPS 1991)
>
> ---
>
> We greatly appreciate your insightful and helpful comments, as they will undoubtedly help us improve the quality of our manuscript. If our response has successfully addressed your concerns, we respectfully hope that you consider raising the score. Should you have any further questions, we would be delighted to engage in further discussion.

---

> > ### Comment · Reviewer_xfGv · 2024-11-24
> >
> > I thank the authors for thoroughly addressing my comments. The new results presented in Table Re1 and Table Re2 resolve my concerns regarding the proposed approach. Additionally, Table Re1, shared in response to reviewer Hu9o, offers a valuable finding regarding the practical utility of the method. Considering all new experiments and clarifications, I am raising my score to 8. I believe that the paper makes a valuable contribution to the field of computational mass spectrometry by developing a method that effectively combines two main MS/MS annotation paradigms.

---

> > > ### Author Response · Authors · 2024-11-25
> > > **Response to Reviewer xfGv**
> > >
> > > Dear Reviewer xfGv,
> > >
> > > Thank you for your positive feedback and continued support for our paper! We appreciate your thoughtful review and are glad that we have adequately addressed your concerns.
> > >
> > > Best regards,
> > >
> > > Authors.

---

### Official Review · Reviewer_eX4t · 2024-11-02

**Soundness:** 3
**Presentation:** 4
**Contribution:** 3
**Rating:** 6
**Confidence:** 4

**Summary:**

The paper introduces a novel approach for predicting peptide sequences from tandem mass spectra, achieving significant performance improvements with a simple retrieval-augmented generation method.

**Strengths:**

**Originality**. The paper proposes an original idea of connecting database search with *de novo* generation for predicting peptide sequences from mass spectra.

**Quality**. The experiments are sound and the source code is provided.

**Clarity**. The paper is well-written and easy to follow.

**Significance**. The performance improvement upon prior work is substantial.

**Weaknesses:**

**Major Comments**

- Missing related work. The paper does not discuss related work on retrieval-augmented generation (RAG) in the context of large language models (LLMs). Is the proposed method novel compared to recent advances in natural language processing?
- Lack of comparison with database search methods. Although overcoming the limitations of database search is a primary motivation, the proposed method is not compared against database search approaches. At least a simple search baseline against the same training database should be implemented for comparison. Ideally, state-of-the-art database search methods should aslo be compared against.
- Potential data splitting issues. The leave-one-out data split by species, while seemingly reasonable, might lead to data leakage. Specifically, the same peptides could appear across different species (potentially even with identical or highly similar MS² spectra). Please analyze the overlap of spectra and peptides between the training and test sets, for example, by assessing vector similarity between binned spectra and sequence identity to evaluate spectral and peptide similarity, respectively.

**Minor Comments**

- Figure 1: Clarify what is meant by “identification rate”—is it the accuracy of preptide prediction?
- Table 1: Confidence intervals or standard deviations for the reported values are missing.
- Line 160: What is the charge state of the precursor ion? Why is it importnant?
- Line 163: “sets” -> “ordered sets of the same size” for mathematical soundness?
- Equation 2: Typo: the comma at the end.
- Line 196: Please clarify how the described features help to attend to differences.
- Line 201: A citation for teacher-forcing is missing.
- Equation 9: Adding one or two sentences explaining the intuition behind this equation could enhance clarity.
- Section 4.1: References for the datasets are missing.
- Section 4.4: If “ReNovo with Residual” is the best-performing method, please clarify why the residual component is not described in the methods section and not used in the final evaluation in Table 1.
- Table 5: Please explain the meaning of “+15.99.”
- Appendix A: References are missing.

**Questions:**

Please see Weaknesses.

---

> ### Author Response · Authors · 2024-11-19
> **Response to Reviewer eX4t (1)**
>
> Thank you for the valuable feedback. I will address each of your concerns in detail.
>
> > W1: Missing related work. The **paper does not discuss related work on retrieval-augmented generation (RAG)** in the context of LLMs. **Is the proposed method novel compared to recent advances in natural language processing?**
>
> Thanks for your insightful reviews!
>
> It must be acknowledged that RAG has demonstrated SOTA performance in numerous generative tasks. The ReNovo model mentioned in this paper draws on an important concept from RAG: enhancing the model's generative capabilities by explicitly retrieving existing information during inference rather than relying solely on the knowledge implicitly stored in the model's parameters.
>
> However, I believe that **the proposed method, ReNovo, is novel compared to recent advances in NLP.** The retrieval-based inference in ReNovo differs significantly from RAG models in the following ways:
>
> - In LLMs, RAG typically involves dynamically integrating up-to-date external information from across the web, effectively incorporating external data. In contrast, **the information retrieved by ReNovo originates solely from the data used during the training stage**, rather than from any external, domain-specific, or test dataset-related sources. This approach serves two purposes: first, to ensure a fair comparison with other de novo baselines, and second, to simulate real-world de novo applications—namely, directly interpreting mass spectra to infer peptides independently of any external peptide database. This independence is crucial in scenarios where peptide databases are unavailable.
>
> - **Our downstream task focuses on de novo peptide sequencing**, which is fundamentally different from the typical tasks of RAG (such as QA or dialogue), given the distinct data formats involved—mass spectra versus text/images. RAG methods cannot be directly applied to MS2 data. Therefore, the way ReNovo processes information (see Section 3.1 ~ 3.2 in paper) is significantly different.
>
> **I believe that the proposed ReNovo method is novel compared to recent advances in RAG, and the RAG methods used in LLMs cannot be directly applied to de novo peptide sequencing, as no applicable external database exists**. Given these significant differences, we did not mention RAG in the original paper.
>
> To aid readers in comprehending our works and further clarify the novelty of our paper, **we have added a section discussing related RAG research in the revised version of the paper**. This section is now available in Section 2.3 of the updated manuscript.
>
> > W2: **Lack of comparison with database search methods.** Although ...
>
> Thanks for your to-the-point reviews!
>
> What we want to emphasize is that **database search methods and de novo methods represent two fundamentally different paradigms in peptide identification, each suited for distinct scenarios. It is unnecessary to conduct a direct performance comparison under identical conditions**. Mainstream database search methods are primarily used to identify known peptides and rely on pre-established large external databases, whereas de novo methods can identify novel peptides that are not included in any database. Such situations include the sequencing of antibodies and novel antigens[1, 2], and the sequencing of metaproteomes lacking established databases[3].
>
> In the experiment of our original paper, the peptide sequences in the test set are entirely distinct from those in the training set. Furthermore, we did not utilize any large-scale, or external databases. Therefore, **database searching methods are not suitable for the de novo peptide sequencing scenarios and datasets presented in our work**.
>
> Additionally, **none of the de novo models [4, 5, 6, 7] have conducted experimental performance comparisons with database searching methods. I believe that it's both common and reasonable not to compare de novo with database search methods when evaluating the performance of de novo approaches**.
>
> ---
>
> [1] Comprehensive evaluation of peptide de novo sequencing tools for monoclonal antibody assembly. (Beslic et al., BIB 2023)
>
> [2] Uncovering thousands of new peptides with sequence-mask-search hybrid de novo peptide sequencing framework. (Karunratanakul et al., Molecular & Cellular Proteomics 2019)
>
> [3] Metaproteomics: harnessing the power of high performance mass spectrometry to identify the suite of proteins that control metabolic activities in microbial communities. (Analytical chemistry 2019)
>
> [4] De novo peptide sequencing by deep learning (Tran et al., PNAS)
>
> [5] Computationally instrument-resolution independent de novo peptide sequencing for high-resolution devices (Qiao et al., NMI)
>
> [6] De Novo Mass Spectrometry Peptide Sequencing with a Transformer Model (Yilmaz et al., ICML 2022)
>
> [7] AdaNovo: Towards Robust De Novo Peptide Sequencing in Proteomics against Data Biases (Jun et al., NeurIPS 2024)
>
> ---
>
> **The next part of our response can be found in the next block.**

---

> ### Author Response · Authors · 2024-11-19
> **Response to Reviewer eX4t (2)**
>
> Thank you for the valuable feedback. I will address each of your concerns in detail.
>
> > W3: **Potential data splitting issues**. The leave-one-out data split by species, while seemingly reasonable, might lead to data leakage. ... **Please analyze the overlap of spectra and peptides between the training and test sets**, for example, by ...
>
> Another good point!
>
> First and foremost, it should be emphasized that the **"leave-one-out data split by species" setting is widely used in de novo methods [1,2,3,4], and we adhere to the same settings**. Additionally, all baselines in our experiments follow the same data split approach to ensure fairness.
>
> **For spectra overlap**, we referred to the method in [5]: Specifically, all spectra in the dataset were discretized into binned spectra. We then calculated the average similarity between each binned spectrum in the test set and all those in the training set. Finally, we averaged these average similarity values across all binned spectra in the testing set to obtain the overall dataset similarity (ranging from [-1, 1], similar to cosine similarity), which **measures the overlap between the training and test spectra**. The results are shown in **Table Re1** below.
>
> **Table Re1: Similarity of Binned Spectra Between Test Set and Training Set.**
> |     | Seven-species Dataset    | Nine-species Dataset | HC-PT Dataset  |
> |-------|:-------:|:-------:|:-------:|
> | Similarity | 0.172 | 0.101 | 0.119 |
>
> From Table Re1, it can be observed that the spectra similarity for all three datasets is approximately 0.1. **This indicates that the overlap between the spectra in the training set and the test set is quite low**.
>
> **For peptide overlap**, we employ a leave-one-out cross-validation framework in which the peptides in the training set are completely disjoint from those in the test set. In other words, during the preprocessing phase of the dataset, **we ensured that there was no duplication between peptide sequences in the training and test sets**, making the dataset suitable for evaluating de novo methods.
>
> To further address your concerns and explore the model's performance on test sequences that lack similar sequences in the training set, we conducted experiments on the Nine-species Dataset, using AdaNovo (the best baseline for this dataset) and CasaNovo for comparison. We began by calculating the Levenshtein distance (a widely adopted method for measuring the similarity between two strings) for each test sequence against all sequences in the training set. We then identified the minimum Levenshtein distance for each test sequence and filtered out test sequences with a minimum Levenshtein distance less than or equal to 3 and 5, respectively, resulting in **Nine-species Test Dataset (>3) and Nine-species Test Dataset (>5). These datasets, having removed potential peptide-level data leakage**, serve to validate "the model’s improvement on test sequences that lack similar sequences in the training set." We maintained all other experimental settings unchanged, and the results are presented in **Table Re2**.
>
> **Table Re2: Performance Comparison on Filtered Test Datasets.**
> | Model Name   | Test Set  | Peptide Precision   | Peptide AUC | AA Precision   | AA Recall |
> |-------|-------|:-------:|:-------:|:-------:|:-------:|
> | CasaNovo  | Nine-species Test Dataset (>3) | 0.369 | 0.324 | 0.638 | 0.637|
> | AdaNovo | Nine-species Test Dataset (>3) | 0.398 | 0.356 | 0.648 | 0.656 |
> | **ReNovo** | Nine-species Test Dataset (>3) | **0.465** | **0.423** | **0.721** | **0.720** |
> | CasaNovo  | Nine-species Test Dataset (>5) | 0.302 | 0.254 | 0.599 | 0.598 |
> | AdaNovo  | Nine-species Test Dataset (>5) | 0.333 | 0.292 | 0.613 | 0.618 |
> | **ReNovo** | Nine-species Test Dataset (>5) | **0.396** | **0.351** | **0.688** | **0.686** |
>
> From Table Re2, we observe that the performance of both ReNovo and SOTA baseline declines on the filtered test datasets, which is expected as the test set contains fewer similar data to the training set. However, it is also evident that **ReNovo still outperforms the SOTA baseline models significantly on the same filtered test sets**. This clearly indicates that **the improved performance of ReNovo is due to generalization rather than overfitting to the training dataset through data leakage**
>
> ---
>
> [1] De novo peptide sequencing by deep learning (Tran et al., PNAS)
>
> [2] Computationally instrument-resolution independent de novo peptide sequencing for high-resolution devices (Qiao et al., Nature Machine Intelligence)
>
> [3] De Novo Mass Spectrometry Peptide Sequencing with a Transformer Model (Yilmaz et al., ICML 2022)
>
> [4] AdaNovo: Towards Robust \emph{De Novo} Peptide Sequencing in Proteomics against Data Biases (Jun et al., NeurIPS 2024)
>
> [5] Comparing similar spectra: from similarity index to spectral contrast angle (Wan et al., Journal of the American Society for Mass Spectrometry 2002)
>
> ---
>
> **The next part of our response can be found in the next block.**

---

> ### Author Response · Authors · 2024-11-19
> **Response to Reviewer eX4t (3)**
>
> We greatly appreciate your insightful comments. I will address each of your concerns in detail.
>
> > **Minor Comments**: Figure 1: Clarify what is meant by “identification rate”—is it the accuracy of preptide prediction?.
>
> Yes, your understanding is correct. In the context of de novo peptide sequencing, specifically in the application scenario of ReNovo as discussed in this paper, the identification rate indeed refers to the accuracy of peptide prediction by the model.
>
> > **Minor Comments**: Table 1: Confidence intervals or standard deviations for the reported values are missing.
>
> Thanks for your helpful reviews!
>
> Due to time constraints, we trained three ReNovo models independently on each of the three datasets with different random initializations, reporting the standard deviation. We selected two key metrics: peptide precision and peptide-level AUC, and compared our results with SOTA baseline models. The detailed results are presented in **Table Re1** below (the values following the "±" indicate the standard deviation.)
>
> **Table Re1: Performance and Standard Deviation.**
> | Model Name   | Dataset  | Peptide Precision   | Peptide AUC |
> |-------|-------|-------| -------|
> | SOTA baseline | Seven-species Dataset | 0.234 | 0.173 |
> | ReNovo | Seven-species Dataset | **0.2810 ± 0.0138** | **0.2317 ± 0.0146** |
> | SOTA baseline| Nine-species Dataset | 0.517 | 0.469 |
> | ReNovo | Nine-species Dataset | **0.5686 ± 0.0136** | **0.5280 ± 0.0127**|
> | SOTA baseline| HC-PT Dataset | 0.419 | 0.373 |
> | ReNovo | HC-PT Dataset | **0.4769 ± 0.0160** | **0.4456 ± 0.0156** |
>
> > **Minor Comments**: Line 160: What is the charge state of the precursor ion? Why is it importnant?
>
> In a standard peptide identification workflow, proteins are broken down into their constituent peptides through enzymatic digestion. The resulting peptides are then separated via liquid chromatography, producing spectra. These spectra reveal the mass and charge information of the intact peptides (also known as the **precursors**). **The charge state of the precursor ion indicates how many positive or negative charges are present on that ion.** The charge state of the precursor ion is important for several reasons: **(1) The charge state affects the observed mass-to-charge ratio (m/z) of the ion**. Since mass spectrometers measure m/z rather than mass alone, understanding the charge state is crucial. **(2)The charge state influences the fragmentation behavior of the ion**. Different charge states can lead to different fragmentation patterns, which can affect the resulting spectra and, ultimately, the interpretation of peptide sequences.
>
> > **Minor Comments**: Line 163: “sets” -> “ordered sets of the same size” for mathematical soundness?
>
> Your review is accurate and valuable; thank you for your suggestions!
>
> > **Minor Comments**: Equation 2: Typo: the comma at the end.
>
> Your review is accurate and valuable; We have fix it in the revised version of the paper. Thank you for your suggestions!
>
> > **Minor Comments**: Line 196: Please clarify how the described features help to attend to differences.
>
> We utilized transformer [1] as the backbone architecture for ReNovo, treating the MS2 data s as a sequence of peaks input into the transformer. Each peak s_i = (m_i,I_i) is treated as a "word" in an MS2 "sentence" s, where **the m/z value m_i represents the "relative position" of the peak in the MS2 data. Similar to how relative positions were considered in the original transformer** model [1], these embeddings help the model focus on m/z variations between peaks.
>
> > **Minor Comments**: Line 201: A citation for teacher-forcing is missing.
>
> Your review is accurate and valuable. We have included the citation [2] in the revised version of the paper—thank you for your suggestion!
>
> > **Minor Comments**: Equation 9: Adding one or two sentences explaining the intuition behind this equation could enhance clarity.
>
> The intuition behind Equations 9 to 10 is to convert the retrieved K context feature - target amino acid pairs N={(kj,vj),j∈(1,2,…,K)} into a distribution over the amino acid vocabulary by applying a softmax function with temperature T to the negative distances, then aggregating over multiple occurrences of the same vocabulary item. A smaller Euclidean distance d indicates that the retrieved feature is closer to the model output context feature, thus requiring a higher weight.
>
>
> ---
>
> [1] Attention is all you need (Vaswani et al., NeurIPS 2017).
>
> [2] A learning algorithm for continually running fully recurrent neural networks  (Williams & Zipser et al., Neural Computation 1989).
>
> ---
>
> **The next part of our response can be found in the next block.**

---

> ### Author Response · Authors · 2024-11-19
> **Response to Reviewer eX4t (4)**
>
> Thank you for the valuable feedback. I will address each of your concerns in detail.
>
> > **Minor Comments**: Section 4.1: References for the datasets are missing.
>
> > **Minor Comments**: Appendix A: References are missing.
>
> Your review is accurate and valuable. We have included citations for the datasets in the revised version of the paper—thank you for your suggestion! Specifically, the seven-species and nine-species datasets are from [1]. The HC-PT dataset is from [2].
>
> > **Minor Comments**: Section 4.4: If “ReNovo with Residual” is the best-performing method, please clarify why the residual component is not described in the methods section and not used in the final evaluation in Table 1.
>
> As outlined in Section 4.4 and Figure 3 of the original paper, ReNovo and ReNovo with Residual exhibit only minor differences in their inference methods and performance. **We introduce the concepts of "ReNovo w/o Retrieval," "ReNovo," and "ReNovo with Residual" to conduct ablation studies that validate the contribution of each component in ReNovo. Our main focus remains on "ReNovo" itself, while the other terms are merely included to facilitate the understanding of the ablation experiments and are not the focal point of the paper**. Consequently, "ReNovo with Residual" was not described in the methods section. Your review is accurate and valuable; we will clearly differentiate these terms in future versions of the paper—thank you for your suggestion!
>
> > **Minor Comments**: Table 5: Please explain the meaning of “+15.99.”
>
> More precisely, "(+15.99)" is used to modify the preceding "M". The notation "M(+15.99)" represents the oxidation of methionine that can be viewed as a special amino acid.
>
> ---
>
> [1] De novo peptide sequencing by deep learning (Tran et al., PNAS 2017)
>
> [2] De novo peptide sequencing with InstaNovo: Accurate, database-free peptide identification for large-scale proteomics experiments' (Eloff et al.)
>
> ---
>
> **We sincerely appreciate the points you have raised. They have helped us identify issues we had not previously considered, particularly regarding some helpful Minor Comments. Sincerely hope that our response can address your concerns and you consider raising the score of our paper. Thank you once again!**

---

> > ### Comment · Reviewer_eX4t · 2024-11-24
> >
> > I thank the authors for addressing my comments, and I have raised my score accordingly. However, I would like to mention a few concerns that could further improve the quality of the paper:
> > - The authors state, “The ReNovo model mentioned in this paper draws on an important concept from RAG,” but also note, “Given these significant differences, we did not mention RAG in the original paper.” I believe it is crucial to reference prior work on RAG in other domains and clearly highlight the key differences with ReNovo. This comparison would better clarify the novelty of the paper and significantly strengthen its contributions.
> > - Regarding “Table Re1: Similarity of Binned Spectra Between Test Set and Training Set,” the description states, “We then calculated the average similarity between each binned spectrum in the test set and all those in the training set.” This approach is not particularly illustrative, as the table only reports the average distance of a test sample to all training samples. To better assess potential data leakage, it would be more informative to report the minimum distance of test samples to the training set.

---

> > > ### Author Response · Authors · 2024-11-25
> > > **Response to Reviewer eX4t**
> > >
> > > Dear Reviewer eX4t,
> > >
> > > Thank you for your positive feedback and continued support for our paper! We appreciate your thoughtful review and are glad that we have adequately addressed your concerns.
> > >
> > > Best regards,
> > >
> > > Authors.

---

> ### Author Response · Authors · 2024-12-01
> **Response to Reviewer eX4t**
>
> Thank you for the valuable feedback. I will address each of your concerns in detail.
>
> > ... I believe it is crucial to reference prior work on RAG in other domains and clearly highlight the key differences with ReNovo. This comparison would better clarify the novelty of the paper and significantly strengthen its contributions.
>
> To aid readers in comprehending our works and further clarify the novelty of our paper, **we have added a section discussing related RAG research in the revised version of the paper**. This section is now available in Section 2.3 of the updated manuscript.
>
> > ... To better assess potential data leakage, it would be more informative to report the minimum distance of test samples to the training set.
>
> To further validate that ReNovo achieves its performance through generalization rather than data leakage leading to overfitting, we conducted additional experiments to assess the impact of dataset similarity on ReNovo's performance at spectra-level.
>
> We adopted the method described in [1]: specifically, all spectra in the dataset were discretized into binned spectra. We then computed the similarity between each binned spectrum in the test set and all binned spectra in the training set. Subsequently, we conducted experiments on the Nine-species dataset, calculating the highest similarity for each spectrum in the test set and ranking them accordingly. **We then filtered out the top 10%, 20%, and 30% of spectra with the highest similarity scores, resulting in three filtered datasets: Nine-species Dataset (Filter 10%), Nine-species Dataset (Filter 20%), and Nine-species Dataset (Filter 30%). These filtered datasets removed potential spectra-level data leakage**. We then evaluated ReNovo on these three filtered datasets, with the results summarized in **Table Re1**.
>
> **Table Re1: The performance of ReNovo on the filtered datasets.**
> |     | Peptide Precision   | Peptide AUC | AA Precision   | AA Recall |
> |-------|:-------:|:-------:|:-------:|:-------:|
> | Nine-species Dataset (Filter 10%) | 0.575 | 0.539 | 0.771 | 0.770 |
> | Nine-species Dataset (Filter 20%) | 0.577 | 0.542 | 0.771 | 0.770 |
> | Nine-species Dataset (Filter 30%) | 0.575 | 0.542 | 0.770 | 0.769 |
>
> From Table Re1, we observe that the performance of ReNovo shows minimal variation across the three filtered datasets. **This indicates that the improved performance of ReNovo is due to generalization rather than overfitting to the training dataset by spectra-level data leakage.**
>
> ---
>
> [1] Comparing similar spectra: from similarity index to spectral contrast angle (Wan et al., Journal of the American Society for Mass Spectrometry 2002)
>
> ---
>
> **Thank you once again for your thorough, insightful, and constructive reviews! If our responses have addressed your concerns fairly, we respectfully hope you might consider raising your score to support our work. Thank you again for your valuable time and constructive feedback!**

---

### Author Response · Authors · 2024-11-17
**General Response to All the Reviewers**

We extend our gratitude to all reviewers for their insightful and constructive feedback on our manuscript!

We are pleased to hear that the reviewers find that **our paper proposes an novel approach / original idea** (3/4 Reviewers eX4t, xfGv, Hu9o) and **clearly defines the strong motivation** (2/4 Reviewers xfGv, Hu9o); Also, they recognize that **our experimental results are sound and provides a thorough analysis of the results** (2/4 Reviewers eX4t, xfGv), and the **performance improvement is substantial and empirically outperforms SOTA approaches** (3/4 Reviewers eX4t, xfGv, 115j); Additionally, they commend **our work for being well-written / well-structured and easy to follow** (3/4 Reviewers eX4t, xfGv, 115j).

We plan to address all suggestions and questions from the reviewers on a point-by-point basis. We remain actively engaged in dialogues and are dedicated to enhancing the quality of this research. Thank you once more for your invaluable contributions!

---

### Public Comment · ~Himanshu_D1 · 2024-11-25
**Concern of this work**

The performance of ReNovo is **significantly** lower than models such as contranovo (https://arxiv.org/pdf/2312.11584) and casanovo v2 (https://www.nature.com/articles/s41467-024-49731-x). The baselines in this paper are all outdated and have inferior performance.  The statement "state-of-the-art performance" is not valid. This leads to the inconceivable effectiveness of the proposed method.

---

> ### Author Response · Authors · 2024-11-26
> **Response to Himanshu D**
>
> Thank you for your comment. I will address your concerns in detail.
>
> > The performance of ReNovo is significantly lower than models such as contranovo ... and casanovo v2 ...
>
> **The performance differences you described are largely attributable to discrepancies in the training and testing datasets.** For instance, both Casanovo v2 [1] and ContraNovo [2] utilize the expansive MassIVE-KB spectral library (identifier: MSV000081142) as training set. This dataset is not only entirely different from ReNovo's training dataset but also significantly larger, making a direct comparison unreasonable and meaningless.
>
> **Additionally, the test datasets used by these models are entirely different.** For example, all models evaluated in our study adopt the widely-used Nine-species dataset from [3] (identifier: MSV000081382) for benchmarking. In contrast, the Nine-species dataset utilized by ContraNovo and Casanovo v2 belong to entirely different versions (identifier: MSV000090982).
>
> **Therefore, given the substantial differences in training and testing datasets, it is entirely unreasonable and meaningless to directly compare the results from Casanovo v2 and ContraNovo‘s paper with those of ReNovo. In contrast, the experimental section of our study conducted fair comparisons of all baseline models on the same datasets, providing more reliable references.**
>
> **Despite the differences in datasets, I firmly believe that the performance of ReNovo surpasses that of Casanovo v2.** Specifically, the AI-level improvements of Casanovo v2 [1] over Casanovo v1 [4] primarily lie in the following aspects: (1) As previously mentioned, the use of a larger dataset. (2)The addition of a beam search decoding strategy. However, in the experimental section of our study, we reproduced the performance of Casanovo using the official GitHub implementation of Casanovo 4.0, which includes the beam search decoding strategy (default beam size = 5). In other words, **the Casanovo model reproduced in our study is architecturally identical to the Casanovo v2 model.** The results, as shown in Table 1 of the original paper (and in the table below, with performance improvements indicated by "+18%"), demonstrate this.
>
> **Table 1: Performance Comparison Between ReNovo and CasaNovo v2**
> | Model Name | Dataset | Peptide Precision | Peptide AUC | AA Precision | AA Recall |
> |-------|-------|-------|-------|-------|-------|
> | CasaNovo v2 | Nine-species Dataset (MSV000081382) | 0.481 | 0.439 | 0.697 | 0.696|
> | ReNovo | Nine-species Dataset (MSV000081382) | 0.568 (+18%) | 0.528 (+20%) | 0.770 (+10%) | 0.769 (+10%)  |
>
> In other words, **the experimental section of our study has already conducted a fair comparison between Casanovo v2 and ReNovo under identical experimental setups, datasets, and data partitioning methods. The results clearly show that the performance of the ReNovo model significantly surpasses that of the Casanovo v2 model.**
>
> **Regarding ContraNovo, the nine-species dataset** (referred to in the original paper as the 9-species-V1) **suffers from a data leakage issue**, where peptide sequences in the test set may also appear in the training set. Specifically, the ICML version of Casanovo [4] states in Section 4.2:
>
> *“To illustrate this point, among the ~26,000 unique peptide labels associated with the human spectra in the test data, 7% overlap with the ~250,000 unique peptide labels associated with spectra from the other eight species.”*
>
> In contrast, the nine-species dataset used in our study underwent strict preprocessing to ensure that the training and test sets contain no overlapping peptide sequences. In other words, **the evaluation datasets employed in our work are much more challenging and widely-used compared to those used in ContraNovo you mentioned.**
>
> More critically, **the motivation behind our dataset choice is that using a dataset containing no duplicate sequences adheres to a key principle of de novo sequencing**: the direct interpretation of mass spectra to infer entirely novel peptide sequences that do not currently exist in any database, which is invaluable in scenarios where peptide databases are unavailable. **In other words, the dataset usage and partitioning methods in this study are more aligned with the real-world application scenarios of de novo models.**
>
> ---
>
> [1] Sequence-to-sequence translation from mass spectra to peptides with a transformer model (Yilmaz et al., Nature Communications)
>
> [2] ContraNovo: A Contrastive Learning Approach to Enhance De Novo Peptide Sequencing (Jin et al., AAAI)
>
> [3] De novo peptide sequencing by deep learning (Tran et al., PNAS)
>
> [4] De Novo Mass Spectrometry Peptide Sequencing with a Transformer Model (Yilmaz et al., ICML 2022)

---

> ### Author Response · Authors · 2024-11-26
> **Response to Himanshu D (2)**
>
> > The baselines in this paper are all outdated and ...
>
> The baseline models in this study include both earlier, milestone-setting models (e.g., DeepNovo) and the latest cutting-edge models, such as **HelixNovo [2] published at BIB (April, 2024), CasaNovo [4] published in Nature Communications (July 2024), and AdaNovo [3] published at NeurIPS (October, 2024). These models are by no means "outdated" and were published in well-regarded conferences or journals this year (2024). Furthermore, all these models claim state-of-the-art performance in their original paper, offering significant reference value.**
>
> In terms of timeliness, these works were published later than ContraNovo [1] (presented at AAAI in March 2024), which you mentioned. This demonstrates that our selection of baselines reflects the most recent advancements in the field.
>
> ---
>
> [1] ContraNovo: A Contrastive Learning Approach to Enhance De Novo Peptide Sequencing (Jin et al., AAAI)
>
> [2] Introducing π-helixnovo for practical large-scale de novo peptide sequencing. (Yang et al., Briefings in Bioinformatics 2024)
>
> [3] AdaNovo: Towards Robust \emph{De Novo} Peptide Sequencing in Proteomics against Data Biases (Jun et al., NeurIPS 2024)
>
> [4] Sequence-to-sequence translation from mass spectra to peptides with a transformer model (Yilmaz et al., Nature Communications)

---

> > ### Public Comment · ~COMMENTOKKKK1 · 2024-11-29
> >
> > Some tricks can result in an improvement on a small dataset but not necessarily on a larger dataset. It's like most people will use ImageNet rather than MNIST nowadays. Improvements in small datasets will NOT help biologists get better results from experiments.
> >
> > MassIVE-KB is a useful dataset that contains only human data as I remember. So the problem of leakage of information is negligible on the 9 species dataset except for human and mouse data.
> >
> > Although HelixNovo and AdaNovo are not published for a long time, the performance is not satisfying as they all use a small dataset(Similar to MNIST).

---

> ### Public Comment · ~Himanshu_D1 · 2024-11-30
> **Comment**
>
> Furthermore, Even on this small dataset, this model's performance is much worse than some of the other Sota work that's trained using the same dataset, rating questions weather it would even work on large data volume.

---

### Meta-Review · Area_Chair_NAcH · 2024-12-20

**Metareview:**

This paper present a denovo peptide sequencing methodology which draws inspiration from database search methods. As the authors state, "By constructing a datastore from training data, ReNovo can retrieve information from the datastore during the inference stage to conduct retrieval-based inference, thereby achieving improved performance." This combination of denovo and database search capability places it somewhere between the two distinct approaches. This point is important when considering to which methods this one should be compared.

The papers strengths were around teh innovative combination of denovo approaches with some database lookup ideas. The presentation and clarity was also a strength.

The main weaknesses were about the support for the claim of state-of-the-art performance, data leakage, and context related to RAG.

The paper clearly has good reasons to support acceptance and good reasons against it. The lengthy discussion about state-of-the-art performance seems to have ended in a place where the claims need to be aligned and calibrated. I very much appreciate the challenge in this effort because it's not easy to place work the combined elements of different approaches into one category. Yet, it seems reasonable to state clearly for the reader which methods the current one outperforms and which it does not, letting the reader decide what state-of-the art means for themselves. The authors have offered to make significant changes to address data leakage concerns, context, and the effect of retrieval. There is value in the exploration of "hybrid" methods and the furtherance of the discussion of such methods in the literature and venue.

**Additional Comments On Reviewer Discussion:**

The core of the discussion surrounded the claim of state-of-the-art performance and what the appropriate comparison set for the claim should be. The reviewers raises concerns about the comparison set which the authors addressed by introducing new concepts: "ReNovo w/o Retrieval," "ReNovo," and "ReNovo with Residual" to assess the contribution of each component.

Another concern was around citations of prior work. The prior work on RAG was initially not cited because it was deemed too distant, but added after reviewer comments.

There was a public comment offered by two anonymous posters, "Himanshu D" and "COMMENTOKKKK", during the review period. The tone of the comments was not one that is typical of a review discussion. Unfortunately, these comments were targeted in their criticism and did not benefit from a full dialogue on all of the aspects of the paper with the other reviewers of the paper. The authors have done a commendable job of remaining disciplined, respectful, and evidence-based in their responses to the public comments. The professionalism of the authors is noteworthy and much appreciated.

---

### Decision · Program_Chairs · 2025-01-22

Accept (Poster)